# Immunomic longitudinal profiling of the NeoPembrOv trial identifies drivers of immunoresistance in high-grade ovarian carcinoma

Olivia Le Saux [1,2,3,4], Maude Ardin[1,2], Justine Berthet[1,2,5], Sarah Barrin[5], Morgane Bourhis [6], Justine Cinier[1,2], Yasmine Lounici[1,2], Isabelle Treilleux[7], Pierre-Alexandre Just[8], Guillaume Bataillon[9], Aude-Marie Savoye[3,10], Marie-Ange Mouret-Reynier [3,11], Elodie Coquan[3,12], Olfa Derbel[13], Louis Jeay [14], Suliman Bouizaguen[14], Intidhar Labidi-Galy [15], Séverine Tabone-Eglinger [16], Anthony Ferrari[17], Emilie Thomas[17], Christine Ménétrier-Caux [1,2,5], Eric Tartour [6], Isabelle Galy-Fauroux [6], Marc-Henri Stern [18], Magali Terme[6], Christophe Caux[1,2,5], Bertrand Dubois[1,2,5,19] ✉ & Isabelle Ray-Coquard [2,3,4,19] ✉

PD-1/PD-L1 blockade has so far shown limited survival benefit for high-grade ovarian carcinomas. By using paired samples from the NeoPembrOv randomized phase II trial (NCT03275506), for which primary outcomes are published, and by combining RNA-seq and multiplexed immunofluorescence staining, we explore the impact of NeoAdjuvant ChemoTherapy (NACT) ± Pembrolizumab (P) on the tumor environment, and identify parameters that correlated with response to immunotherapy as a pre-planned exploratory analysis. Indeed, i) combination therapy results in a significant increase in intraepithelial CD8$^+$PD-1$^+$ T cells, ii) combining endothelial and monocyte gene signatures with the *CD8B/FOXP3* expression ratio is predictive of response to NACT + P with an area under the curve of 0.93 (95% CI 0.85-1.00) and iii) high *CD8B/FOXP3* and high *CD8B/ENTPD1* ratios are significantly associated with positive response to NACT + P, while *KDR* and VEGFR2 expression are associated with resistance. These results indicate that targeting regulatory T cells and endothelial cells, especially VEGFR2$^+$ endothelial cells, could overcome immune resistance of ovarian cancers.

High-grade serous carcinoma (HGSC) is the seventh most frequent cancer worldwide and the most common cause of mortality due to gynecological cancers[1]. Despite the fact that intra-tumoral T cells are frequent in advanced HGSC and are associated with improved clinical outcome[2], immune checkpoint inhibitors (ICI), such as PD-1/PD-L1 inhibitors, exhibit modest activity as single agents, with an objective response rate (ORR) of ~10%, and infrequent durable responses[3]. Even

though neoadjuvant chemotherapy (NACT) increases T cell infiltration, especially of CD8$^+$ T cells, in both the stroma and the tumor[4–6], as well as their effector functions[6–8], combining ICI to chemotherapy in first-line or recurrent settings has so far had no benefit for the survival HGSC patients[9–11]. Several biomarkers have been associated with a greater sensitivity to PD-1/PD-L1 inhibitors[9,11], including (high) PD-L1 expression (20-57%) alone or combined with intra-tumoral CD8$^+$ T cells

(35%), but are not sufficient for selecting candidates amenable to PD-1/PD-L1 blockade[12,13].

Aside from T cells, tumor-associated macrophages (TAMs), especially of the M2-like immune-suppressive phenotype[14,15], could also play a role in driving resistance to ICI. Being the most abundant immune subpopulation within HGSC[16] and thus likely to establish frequent communication networks with other cells of the tumor microenvironment[17], they were reported to be involved in decreasing T cell effector functions and/or promoting angiogenesis. Angiogenesis, which is involved in tumorigenesis, tumor growth, and ascites formation in HGSC, has also been associated with immunosuppression through a reduction of effector T cells and an increase in regulatory T cells in the tumor microenvironment[18–21]. A better understanding of the spatial interactions between the various tumor immune cells may unveil targets to overcome immune resistance to ICI.

In this study, we comprehensively analyzed paired tumor samples, before (at diagnosis) and after treatment (during interval debulking surgery, IDS), from patients of the NeoPembrOv randomized phase II trial[22], providing us with the opportunity to longitudinally profile the tumor immune environment, to improve our understanding of the biology of the disease, and to identify putative mechanisms of resistance/sensitivity to ICI. The NeoPembrOv study, which evaluated neo-adjuvant Pembrolizumab in combination with NACT ± bevacizumab in patients with unresectable FIGO stage IIIC/IV HGSC, demonstrated that combining Pembrolizumab to NACT ± bevacizumab was feasible with limited adverse effects and displayed some promising results for a fraction of patients[22]. Here, by combining transcriptomics (bulk RNA sequencing) and spatial characterization of the tumor microenvironment (by in situ multiplexed immuno-fluorescence and digital pathology), we decipher the impact of NACT ± Pembrolizumab on the immune infiltrate and identify prognostic and predictive biomarkers of response or resistance to immunotherapy. In this work, we show that i) NACT + P results in a significant increase in intraepithelial CD8+PD-1+ T cells, ii) combining endothelial and monocyte gene signatures with the *CD8B/FOXP3* expression ratio is predictive of response to NACT + P and iii) high *CD8B/FOXP3* and high *CD8B/ENTPD1* ratios are significantly associated with positive response to NACT + P, while *KDR* and VEGFR2 expression are associated with resistance.

## Results

### RNAseq reveals changes in tumor T cell infiltration following NACT±P

To quantitatively characterize changes in the tumor immune microenvironment induced by neoadjuvant platinum-based chemotherapy ± P, we compared pre- and post-treatment matched tissue samples from 64 patients (n = 23 in the NACT arm and n = 41 in the NACT + P arm, see "Methods" section) with contributive material using bulk RNA-sequencing (RNA-seq) (Supplementary Fig. 1). No significant differences of clinical characteristics or progression were observed between the whole NeoPembrOv cohort and the groups of patients used for multi-IF and RNA-seq (Supplementary Table 1). Immune and non-immune cell populations were inferred using two computational tools, Microenvironment Cell Populations-counter (MCP-counter)[23] and quanTIseq[24]. Expression scores of individual genes and gene signatures in the different samples are presented as a heatmap in Fig. 1A. *PTPRC* (CD45) expression increased significantly in approximately 65%, 95% CI [45−80] of patients in the experimental arm (NACT + P) vs 43%, 95% CI [23−66] (not statistically significant) in the NACT control arm (Fig. 1A, B), indicating that immune cell infiltration preferentially increased when patients received the combination therapy. On the opposite, *EPCAM* expression decreased significantly after treatment in both arms (Fig. 1A, B). After treatment, we observed a significant upregulation of the CD8+ T cell signature ($_{unadj}P$ = 0.001, FDR = 0.07 for MCP-counter and $_{unadj}P$ = 0.031, FDR = 0.079 for quanTIseq) for

patients receiving NACT + P, unlike those receiving NACT only ($_{unadj}P$ = 0.452, FDR = 0.888 and $_{unadj}P$ = 0.816, FDR = 0.932 respectively) (Fig. 1A, B and Supplementary Fig. 2A). A higher fraction of patients treated with NACT + P displayed increased expression of the CD8+ T cell gene signature (MCP-counter) post-treatment (71%, 95% CI [52−85] vs 52%, 95% CI [30−74]) (Fig. 1B). Genes associated with NK cells and plasma cells (*JCHAIN* and *IGHA1-2*) increased significantly in both arms after treatment, suggesting a prime role of chemotherapy in these effects (Fig. 1A, B). Together, these observations suggest that Pembrolizumab increases tumor infiltration with immune cells and CD8+ T cells compared to NACT alone.

Analysis of the differentially expressed genes (DEG) between pre- and post-treatment samples (fold-change > 1.5, adjusted *p*-value < 0.05) revealed a downregulation of pathways related to cell division and DNA replication after either treatment, consistent with a cytostatic effect of NACT, while pathways related to extracellular matrix were enriched after treatment (Fig. 1C). Interestingly, the NACT + P arm alone displayed a significant enrichment in pathways related to T cell activation and differentiation post-treatment (Fig. 1C), suggesting that PD-1 blockade had reinvigorated T cells. To investigate the specific effect of Pembrolizumab, we focused on DEG in the NACT + P arm (Supplementary Fig. 2B). We observed an upregulation of genes related to CD8 T cell infiltration (*CD3D, CD3E, CD8A, CD8B*) and function (*GZMM, GZMK, ICOS, LAG3, TIGIT*), as well as genes associated with B cells/plasma cells (*TNFRSF17, MZB1, IRF4*, Ig heavy and light chain genes), suggesting that Pembrolizumab boosted T and B cell-mediated immune responses (Supplementary Fig. 2B).

Collectively, these data demonstrate that Pembrolizumab amplifies the effect of NACT on the recruitment of T and B cells to the tumor and promotes T cell differentiation and activation.

### The combination of Pembrolizumab and NACT increases intra-epithelial CD8+PD-1+T cells

To further investigate the effect of NACT ± P on T cells, we stained paired tumor samples from the 64 patients with a 7-color immuno-fluorescence panel (CD4, CD8, PD-1, Ki67, ActCasp3, panCK) to analyze changes in tumor cells and T cell subsets upon treatment (Supplementary Fig. 3A). Machine-learning based quantification of digital images revealed that the proportion of apoptotic cells (ActCasp3) in tumor cells was virtually unchanged after treatment, while that of proliferative cells (Ki67) decreased significantly in both treatment arms (Supplementary Fig. 3B), likely reflecting the cytotoxic effect of chemotherapy.

Moreover, the abundance of CD8+, but not CD4+, T cells increased after treatment in a major fraction of patients from both arms, but these changes only reached statistical significance in the stroma for patients receiving chemotherapy alone and in tumor islets for patients receiving the combination (Fig. 2A), suggesting that chemotherapy-induced an influx of CD8+ T cells in the stroma, while adding Pembrolizumab favored their localization to the tumor islets. Emerging evidence indicates that only a minor fraction of tumor-infiltrating T cells are tumor-specific and that bystander T cells recognizing cancer-unrelated antigens are abundant. Considering PD-1 expression, which can be used to delineate Ag-specific cells[25], we observed a significant increase in CD4+PD-1+ T cells in the stroma after NACT, and of CD8+PD-1+ T cells in tumor islets after NACT + P treatments (Fig. 2B). Although the ratio of CD8+PD-1+ T cell density between tumor and stromal zones was similar between both arms before treatment, it significantly increased in the NACT + P arm after treatment (Fig. 2C). These results corroborate micrographs of a patient illustrated in Fig. 2D, in which CD8+PD-1+ T cells mainly located in the stroma before treatment accumulated in tumor islets after a combined NACT + P therapy. In addition, the median distance between an apoptotic tumor cell (activated caspase-3 positive) and the nearest CD8+PD-1+ T lymphocyte significantly decreased after NACT + P, but not NACT,

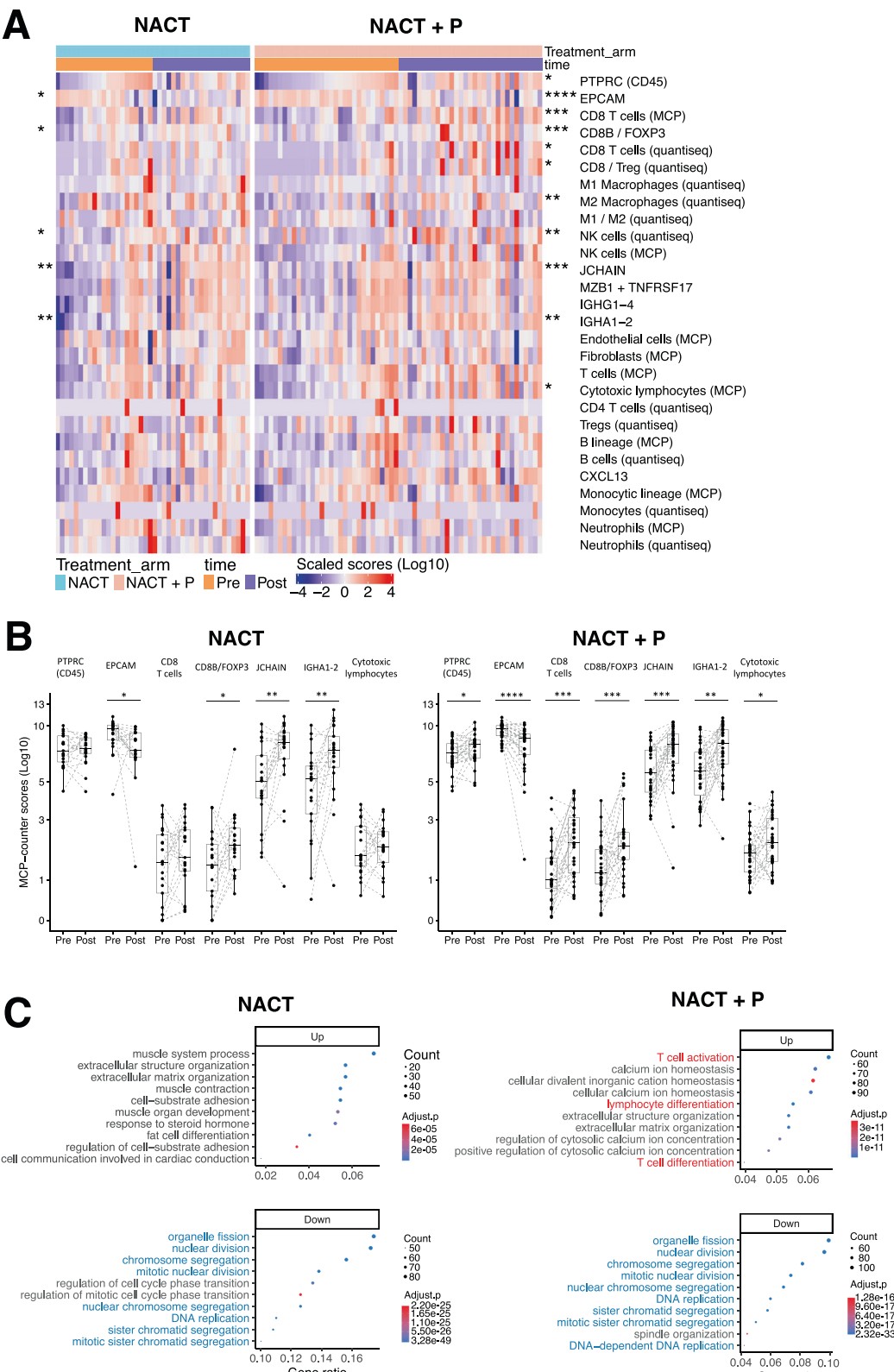

treatment (Supplementary Fig. 3C). No significant difference between both arms was observed for other immune cells (Fig. 2B). Furthermore, tumors with an increased density of intraepithelial CD8+PD-1+ T lymphocytes after treatment were associated with higher neoantigen-specific tumor-infiltrating lymphocyte signature scores[26], suggesting that at least part of these effector cells were tumor-specific (Supplementary Fig. 3D). Further characterization of T cells according to their

proliferation status (Ki67) did not reveal any difference between pre- and post-treatment samples of either arm of the trial (Supplementary Fig. 3E). Although a high density of CD8+ PD-1+ T cells in tumor islets after NACT + P was not significantly associated with increased progression-free survival (PFS) ($p = 0.13$, 95% CI [0.3–1.2]), it tended to be associated with improved overall survival (OS; $p = 0.08$, 95% CI [0.2–1.1]; Fig. 2E).

**Fig. 1 | Impact of neoadjuvant chemotherapy ± Pembrolizumab on the HGSC microenvironment using bulk RNAseq. A** Heatmap depicting the expression level of selected individual genes, gene signatures (MCPcounter and quanTIseq) and gene ratios inferred from bulk RNAseq data of tumors collected before (orange) or after treatment (purple) from patients of the NeoPembrOv trial receiving NACT (blue) or NACT + P (red). Red and Blue colors represent a high and low expression score, respectively. Statistically significant unadjusted *p*-values from two-sided Wilcoxon signed rank test between pre- and post-treatment samples are indicated on the left for the control arm and on the right for the experimental arm. Patients (x-axis) are ordered according to increasing *PTPRC* (CD45) expression before treatment and are listed in the same order for post-treatment samples. **B** Box and dotplots depicting the MCP counter gene signature expression scores in pre- and post-treatment samples for each patient in the NACT arm (left, *N* = 21 patients) and the NACT + P arm (right, *N* = 31 patients). The centerline of boxes depicts the median values; the bottom and top box edges correspond to the first and third

quartiles. Statistical significance was evaluated using two-sided Wilcoxon signed-rank tests and $_{unadj}$P values are reported for NACT arm: *EPCAM* = 0.016; *CD8B/FOXP3* = 0.038; *JCHAIN* = 0.008; *IGHA1-2* = 0.004 and NACT + P arm: *PTPRC* (CD45) = 0.027; *EPCAM* = 8.66e-05; CD8 T cells = 0.001; *CD8B/FOXP3* = 0.001; *JCHAIN* = 0.001; *IGHA1-2* = 0.002; Cytotoxic lymphocytes = 0.027. **C** Graphs depicting the top 10 pathways (GO BP) enriched in the differentially expressed genes between post- and pre-treatment samples for each treatment arms (upregulated pathways are in the top and down regulated are in the bottom part of the figure). Pathways highlighted in blue are common to both arms (NACT and NACT + P). Pathways highlighted in red are specific to the NACT + P arm. Only genes with a fold-change > 1.5 and an adjusted *p*-value < 0.05 were considered. Gene ratio represents the percentage of differentially expressed genes (DEG) identified in the pathway, dot size (Count) is representative of the number of DEG considered for pathway analysis and adjusted *p*-values (adjust.p) were obtained from one-sided Fisher exact test. Source data are provided as a Source Data file.

Together, our data demonstrate that NACT increased tumor-infiltrating CD8$^+$ T cells, and that the addition of Pembrolizumab promoted localization of CD8$^+$PD-1$^+$ T cells in the vicinity of tumor cells.

## Combining endothelial and monocyte gene signatures and the *CD8B/FOXP3* expression ratio is predictive of response to NACT+P

As patients generally receive Pembrolizumab for a duration of 24 months, we investigated the number of progressors (Pr) vs non-progressors (NPr) under Pembrolizumab at 24 months. This landmark is also consistent with the median PFS reported in the French Anthalya trial[27]. This criterion was favored over the completeness of cytoreduction score (CC-score) at interval debulking surgery, as survival endpoints capture responses to immunotherapeutic agents better than the objective response rate or equivalent measures such as the CC-score[28,29]. To identify predictive biomarkers of non-progression under NACT + P, pre-treatment gene signature expression scores for immune and non-immune populations, inferred from bulk RNAseq data, and pre-treatment densities of various T and B cell populations, calculated based on two 7-color multi-IF staining panels (Supplementary Figs. 3A and 4A), were compared between Pr and NPr. A univariate analysis revealed a statistically significant increased expression of the *CD8B/FOXP3* gene ratio exclusively in NPr in the NACT + P arm at 24 months (Fig. 3A and Supplementary Fig. 4B). The gene signature for monocytes was positively and significantly associated with response in the NACT arm with the same trend in the NACT arm although not reaching statistical significance (Fig. 3A and Supplementary Fig. 4B). Conversely, the gene signature for M2-like macrophages was negatively associated with response to NACT + P, with a similar trend for patients receiving NACT alone (Fig. 3A Supplementary Fig. 4B). In addition, gene expression signature scores for endothelial cells and neutrophils were significantly negatively associated with response to NACT + P (Fig. 3A abd Supplementary Fig. 4B). None of the immune parameters measured by multi-IF, including densities of T cell subsets, B cells, IgG and IgA plasma cells and TLS, were associated with response in either treatment arms (Supplementary Fig. 4C, D).

To select significant and independent biomarkers associated with response to NACT + P, we used a multivariate logistic regression model with the least absolute shrinkage and selection operator (LASSO) on patients treated with NACT + P. We included in the model all pre-treatment biomarkers significantly associated with response to NACT + P in univariate analyses, e.g., MCP counter endothelial and neutrophil signature scores, the *CD8B/FOXP3* ratio and the quanTIseq monocyte and M2-like macrophage signature scores. By considering other potential predictive biomarkers in HGSC (PD-L1 combined positive expression score (CPS) and intra-epithelial CD8$^+$ cell density at baseline (IF)) and by categorizing variables using

known thresholds[9,30] or optimal cut-offs, we found that gene signature scores for monocytes and for endothelial cells, as well as the *CD8B/FOXP3* expression ratio, were independent predictive factors associated with response. To limit overinterpretation and overfitting, 100 bootstrap samples were generated and only variables selected through all the 100 corresponding lasso Cox models were retained in the final model (Table 1). Importantly, the endothelial cell signature showed a significant interaction with treatment arm and the *CD8B/FOXP3* ratio a trend towards a significant interaction (*P* = 0.021 and 0.060, respectively), suggesting that the endothelial signature was a predictive biomarker of response. The monocyte signature showed no interaction with treatment arm (*p* = 0.559) (Table 1). Since the experimental arm is not an ICI-only arm, we included in the model parameters that interact with the treatment and are thus linked to pembrolizumab. We also considered parameters without interaction that are related to NACT. The area under the curve (AUC) for the prediction model considering these 3 parameters was 0.93 (95% CI [0.85–1.00]), highlighting excellent discrimination performance (Fig. 3B). In comparison, for the prediction models including PD-L1 expression alone or in combination with intra-epithelial CD8$^+$ T cells, the AUC were 0.60 [0.42–0.78] with a 95% CI and 0.63 [0.44–0.82], respectively (Fig. 3B). Interestingly, all models gave a similar AUC in the NACT alone arm, confirming that our model indeed predicted response to the combined treatment and not to chemotherapy alone (Fig. 3B).

## The monocyte signature, as well as *TREM2* and *SMARCD3* expression are associated with survival irrespective of treatment arm and are thus prognostic factors

Among the three factors included in our prediction model, the pre-treatment monocyte signature did not show a specific interaction with the treatment arm, suggesting that it is a prognostic biomarker rather than a predictive biomarker of response to Pembrolizumab (Table 1). To further investigate the prognostic impact of this signature, we analyzed the outcome of patients regardless of the therapeutic arm. A high (positive) monocyte signature score was associated with a better median PFS (38 vs 18 months) with a clinically relevant hazard ratio (HR) of 0.45 [95% CI 0.19–0.95] and a trend towards a better OS (not reached vs 36 months) (HR = 0.33 [95% CI 0.07–1.01]) (Fig. 4A). Despite a common ontogeny, monocytes and M2-like macrophages displayed opposite associations with treatment response in univariate analysis and were negatively correlated (Fig. 4B). To confirm these associations, we analyzed specific genes of individual monocyte/macrophage cell subsets. Among the top 5 genes differentiating the 2 cell subtypes, *TREM2*, a member of the Triggering Receptor Expressed on Myeloid cells (TREM) family, was more than 8-fold upregulated in M2-like macrophages compared to M1-like macrophages, and was largely not expressed by monocytes (<1 TPM; Supplementary Table 2). Conversely, *SMARCD3*, a gene

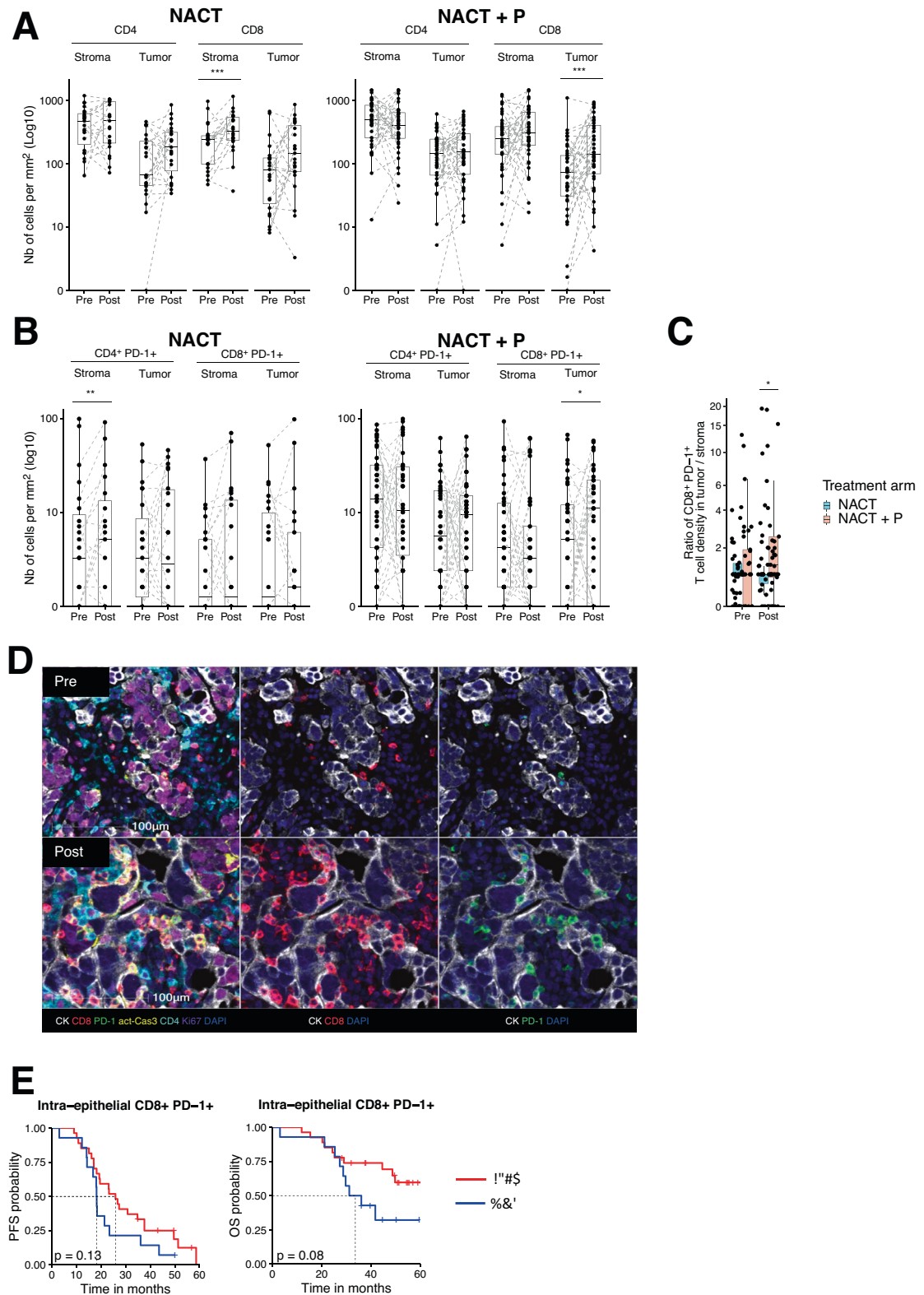

specifically enriched in monocytes[31], was one of the top 5 genes of the quanTIseq monocyte signature and was poorly expressed by M2-like macrophages (Supplementary Table 3). *TREM2* expression was associated with a shorter PFS, although not reaching statistical significance, and with a significantly shorter OS (Fig. 4C). Conversely, *SMARCD3* expression was associated with a significantly longer PFS and OS (Fig. 4D). These results indicate that both SMARCD3 and TREM2 could be targeted to enhance prognosis in HGSC.

## A high *CD8B/FOXP3* expression ratio is associated with better survival for patients treated with NACT+P

The second most influential factor in our predictive model was the *CD8B/FOXP3* ratio (Table 1). Indeed, a high expression of *CD8B/FOXP3* ratio (>1.5) was associated with prolonged PFS in the NACT + P arm compared to the NACT arm (median PFS of 38 vs 20 months, HR = 0.42 [95% CI 0.18–0.98], $p = 0.046$), and OS (median OS not reached vs 35 months, HR = 0.33 [95% CI 0.11–0.90], $p = 0.04$) (Fig. 5A). *CD8B/*

**Fig. 2 | Effect of neoadjuvant chemotherapy +/- Pembrolizumab on T cells in HGSC revealed by multiplex IF tissue imaging. A** Box and dotplots representing the density (number of cells/mm²) of CD4⁺ and CD8⁺ cells in the tumor versus stroma in pre- and post-treatment samples of patients receiving NACT (left, $N = 21$ patients) vs NACT + P (right, $N = 31$ patients). The centerline of boxes depicts the median values; the bottom and top box edges correspond to the first and third quartiles. Statistical significance was evaluated using two-sided Wilcoxon signed-rank test. $_{unadj}P$-values are reported. NACT: CD8 Stroma, $_{unadj}p = 8.49$e-04. NACT + P: CD8 Tumor, $_{unadj}p = 7.90$e-04. **B** Box and dotplots representing the number of cells/mm² (i.e., the density) of CD4⁺PD-1⁺ and CD8⁺PD-1⁺ cells both in the tumor and stroma pre- and post-treatment in the NACT arm (left, $N = 21$ patients) and the NACT + P arm (right, $N = 31$ patients). The centerline of boxes depicts the median values; the bottom and top box edges correspond to the first and third quartiles. Statistical significance was evaluated using two-sided Wilcoxon signed-rank test. $_{unadj}P$-values are reported. NACT: CD4⁺ PD-1⁺ Stroma; $_{unadj}p = 2.64$e-03.

NACT + P: CD8⁺ PD-1⁺ Tumor, $_{unadj}p = 4.11$e-02. **C** Box and dotplots representing the ratio of CD8⁺PD-1⁺ cell density in tumor / CD8⁺PD-1⁺ cell density in stroma in pre- and post-treatment samples in patients receiving NACT (blue, $N = 21$ patients) and patients receiving NACT + P (red, $N = 31$ patients). The centerline of boxes depicts the median values; the bottom and top box edges correspond to the first and third quartiles. Statistical significance was evaluated using two-sided Wilcoxon rank sum test. Post: $_{unadj}p = 2.81$e-02. **D** Representative 7-color multiplex IF images of a tumor sample collected at baseline (pre-treatment, top) and at interval debulking surgery (IDS) (post-treatment, bottom) from patients receiving NACT + P showing an increase in intra-epithelial CD8⁺PD-1⁺ cells. The composite image is shown on the left while selected channels are shown on the right. **E** PFS (left) and OS (right) curves according to the intra-epithelial CD8⁺PD-1⁺ density after treatment for patients in NACT + P arm. Patients were stratified based on the best cutoff (High (red), $n = 27$; Low (blue), $n = 14$). Statistical comparison of survival curves was performed using the likelihood ratio test. Source data are provided as a Source Data file.

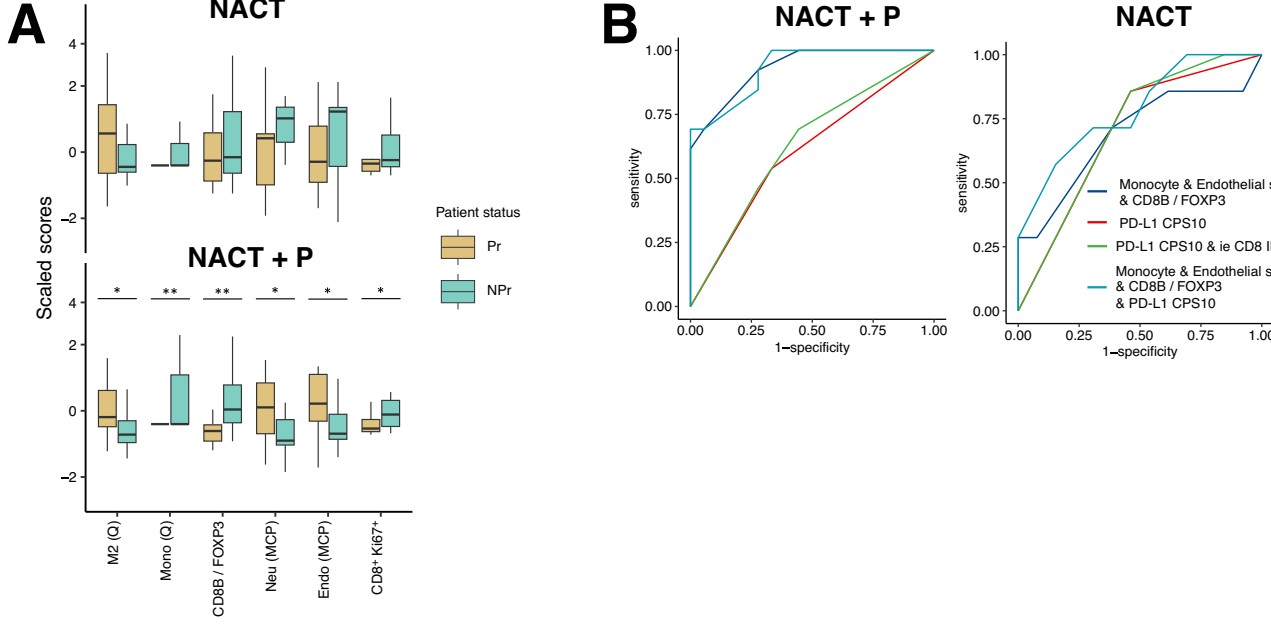

**Fig. 3 | Predictive biomarkers of response to NACT + Pembrolizumab.**
**A** Boxplots representing the scaled expression scores of RNAseq gene signatures before treatment and of immune cell densities before treatment that significantly differed between non progressors, NPr, (green-blue, $N = 7$ NACT and $N = 13$ NACT + P) and progressors, Pr, (dark yellow, $N = 13$ NACT and $N = 19$ NACT + P) receiving NACT (top) versus patients receiving NACT + P (bottom). The centerline of boxes depicts the median values; the bottom and top box edges correspond to the first and third quartiles. Statistical significance was evaluated using two-sided Wilcoxon rank sum test and unadjusted $p$-values are indicated on top of boxplots.

For NACT + P arm, M2 (Q) = 0.037; Mono (Q) = 0.006; *CD8B/FOXP3* = 0.008; Neu (MCP) = 0.045; Endo (MCP) = 0.027; CD8⁺Ki67+ = 0.041 **B** ROC curves representing the area under curve for the models considering monocyte & endothelial gene signature expression & *CD8B/FOXP3* gene expression ratio (dark blue), the same 3 variables and PD-L1 expression (light blue), PD-L1 expression and intra-epithelial CD8⁺ T cell density at baseline (green) and PD-L1 expression alone (red) in the NACT + P arm (left) vs NACT (right) arm. 95% CI was calculated for the area under the curve of the different models. No adjustment for multiple comparisons was made. Source data are provided as a Source Data file.

**Table 1 | LASSO logistic regression model**

| Factor | β | Odds Ratio | SE | P-value | Interaction × treatment arm |
|---|---|---|---|---|---|
| Monocyte sig (Q) | −0.205 | 0.815 | 0.348 | 0.783 | *P = 0.559* |
| CD8B/FOXP3 ratio | +0.523 | 1.687 | 0.272 | 0.223 | *P = 0.060* |
| Endothelial sig (MCP) | −0.656 | 1.927 | 0.274 | 0.008 | *P = 0.021* |

Results from the multivariate LASSO logistic regression model using all significant variables from panel A and predictive biomarkers reported in the literature. Only variables selected by the LASSO method are listed. β is the coefficient or estimate quantifying the potential contribution of each independent variable to the response. Negative and positive values indicate a negative and positive association with the response to treatment. Coefficients represent the log odds ratio for response. Source data are provided as a Source Data file.
*SE* standard error.

*FOXP3* expression ratio and intra-tumor CD8⁺ T cell density were positively correlated, indicating consistency between transcriptomic and proteomic data (Supplementary Fig. 5A). No major differences in PFS and OS were observed according to pre-treatment intra-tumor CD8⁺ T cell density (Supplementary Fig. 5B). We also analyzed the predictive value of other Treg-associated genes and found that the

ratios of *CD8B* to *ENTPD1* (CD39), *HAVCR2* (TIM-3) or *TIGIT* were also associated with response without reaching statistical significance ($_{unadj}P < 0.10$) (Supplementary Table 4). All three markers were correlated with *FOXP3* expression (Fig. 5B and Supplementary Fig. 5C, D). In addition, flow cytometry (FCM) analysis of CD39 (encoded by *ENTPD1*) in HGSC samples revealed a greater expression in tumor-infiltrating

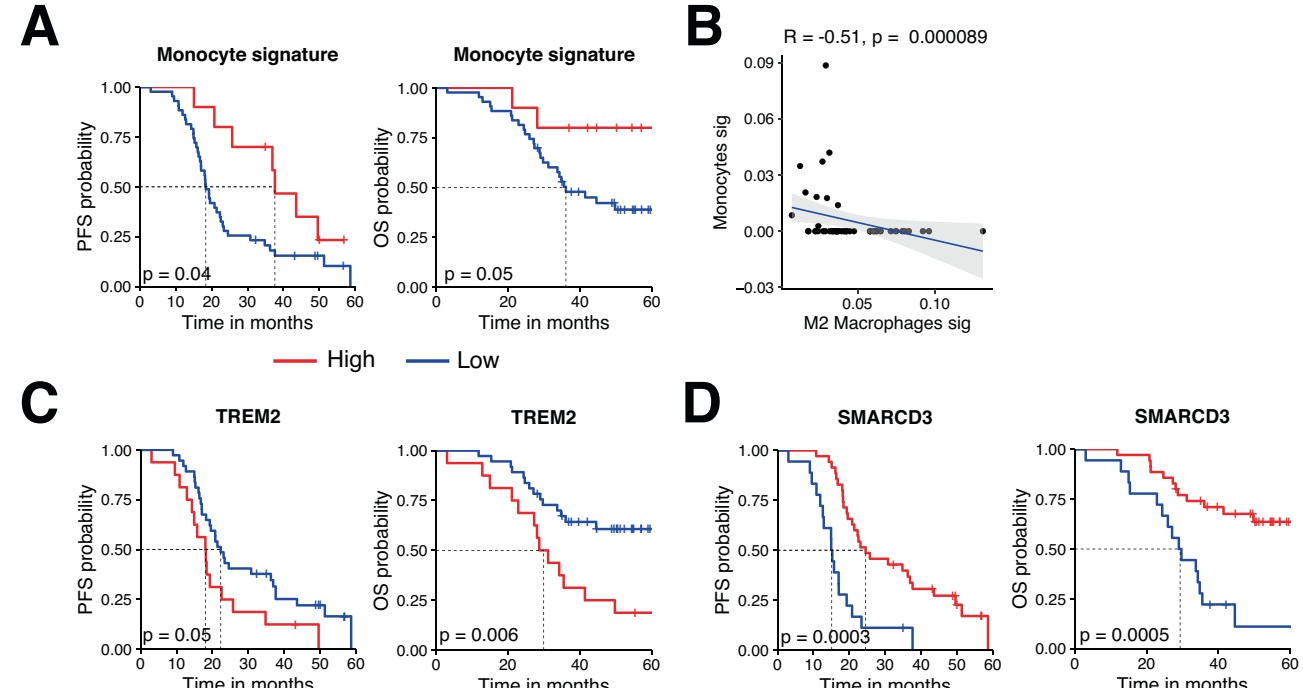

**Fig. 4 | The monocyte signature, a low *TREM2* expression and a high *SMARCD3* expression are associated with survival irrespective of treatment arm and are thus prognostic factors. A**, **C**, **D** PFS (left) and OS (right) curves of patients included in the NeoPembrOv trial (without distinction of treatment arm) according to the expression of a monocyte gene signature (High (red), n = 10; Low (blue), n = 43) (**A**), TREM2 gene expression (High (red), n = 16; Low (blue), n = 37) (**C**) and SMARCD3 gene expression (High (red), n = 35; Low (blue), n = 18) (**D**). Patients were stratified based on a positive (=high) versus negative (=low) expression score (**A**), and on the best cut-off (**B**, **C**). Statistical comparison of survival curves was performed using the likelihood ratio test. **B** Spearman correlation between monocyte and type 2 macrophage expression signature. Error bands represents the 95% CI as a shaded gray area. Source data are provided as a Source Data file.

Tregs compared to CD4$^+$ and CD8$^+$ effector T cells (Supplementary Fig. 5E), indicating that Tregs may account in a large part for *ENTPD1* expression. Interestingly, a strong *CD8B/ENTPD1* expression ratio (Fig. 5C) was associated with better overall survival in the NACT + P arm compared to NACT (median PFS of 33 vs 19 months, respectively, HR = 0.53 [95% CI 0.24–1.20], p = 0.13; median OS NR vs 34 months, respectively, HR = 0.36 [95% CI 0.13–0.97], p = 0.05). This was not the case for *CD8B/TIGIT* and *CD8B/HAVCR2* (Supplementary Fig. 5F, G). Altogether, these data suggest that the ratio of CD8$^+$ T cells to CD39-expressing Tregs positively influences the response to NACT + P and that CD39 may constitute a therapeutic target to overcome resistance to Pembrolizumab.

### High *KDR*/VEGFR2 expression ratio is associated with resistance to NACT+P

The pre-treatment endothelial signature was the only factor independently and significantly associated with resistance to NACT + P treatment. To evaluate the predictive impact of this endothelial cell gene signature variable, we categorized patients into "high" and "low" expressors using the optimal threshold, resulting in 62.3% vs 37.7% of patients having a high or low gene expression signature score, respectively. A clear interaction between the expression level of the endothelial cell gene signature and PFS (p = 0.003, 95% CI [0.04-0.56]), and OS (p = 0.02, 95% CI [0.04-0.77]) was observed, when patients were treated with NACT + P (Fig. 6A). As the fraction of patients that did not receive Bevacizumab was higher in the NACT + P arm compared to the NACT arm, albeit not reaching statistical significance, (14.8% vs 6.67% respectively, p = 0.328), we ran the same analysis exclusively on patients who had received the antiangiogenic (Supplementary Fig. 6A). Similar results were obtained, indicating that the endothelial signature was associated with resistance to Pembrolizumab but not Bevacizumab. Moreover, no difference was

observed between progressors and non-progressors with regards to the use of Bevacizumab (87.0% vs 91.7% respectively, p = 0.734).

Next, to further identify potential targets associated with resistance to NACT + P, we analyzed each gene of the endothelial MCP signature for its association with response (Supplementary Fig. 6B). Among the genes significantly overexpressed in progressors compared to non-progressors was *KDR* (Supplementary Fig. 6B), encoding for VEGFR2, one of the two VEGF receptors. Multiplex-IF tumor tissue staining revealed that VEGFR is expressed by endothelial cells (identified as CD31$^+$ cells, Fig. 6B), as expected. In addition, the endothelial gene signature expression score at baseline was positively correlated with the percentage of CD31$^+$ cells expressing VEGFR2 (p = 0.04) (Fig. 6C). Importantly, a low expression of VEGFR2 on CD31$^+$ endothelial cells was associated with increased survival of patients after treatment with NACT + P (Fig. 6D). To confirm this finding and as no transcriptomic dataset of HGSC patients under ICI was available, we used an external validation cohort of 102 head and neck cancer patients treated with ICI[32]. In this external validation cohort, a high *KDR* expression was also associated with poorer OS (p = 0.04, 95% CI [0.49-0.98]) (Supplementary Fig. 6C). To confirm these findings, we analyzed gene expression of individual VEGFR2 ligands[33]. As PDGF-alpha can heterodimerize with PDGF-beta, we analyzed the sum of *PDGFA* + *PDGFB*. *PDGFA* + *B* expression was negatively associated with response for patients receiving NACT + P (p = 0.001, FDR = 0.009) and this was mainly attributable to *PDGFA* (p = 0.003, FDR = 0.009) (Supplementary Fig. 6D). Kaplan-Meier survival curves showed that there was a statistically significant interaction between the level of *PDGF*A/B expression and both PFS (p = 0.02) and OS (p = 0.002) (Supplementary Fig. 6E) exclusively in patients receiving the combination of NACT + P.

Collectively, these data indicate that resistance to NACT + P is associated with a specific microenvironment characterized by the high expression of *KDR* orVEGFR2 and the angiogenic factor *PDGF*.

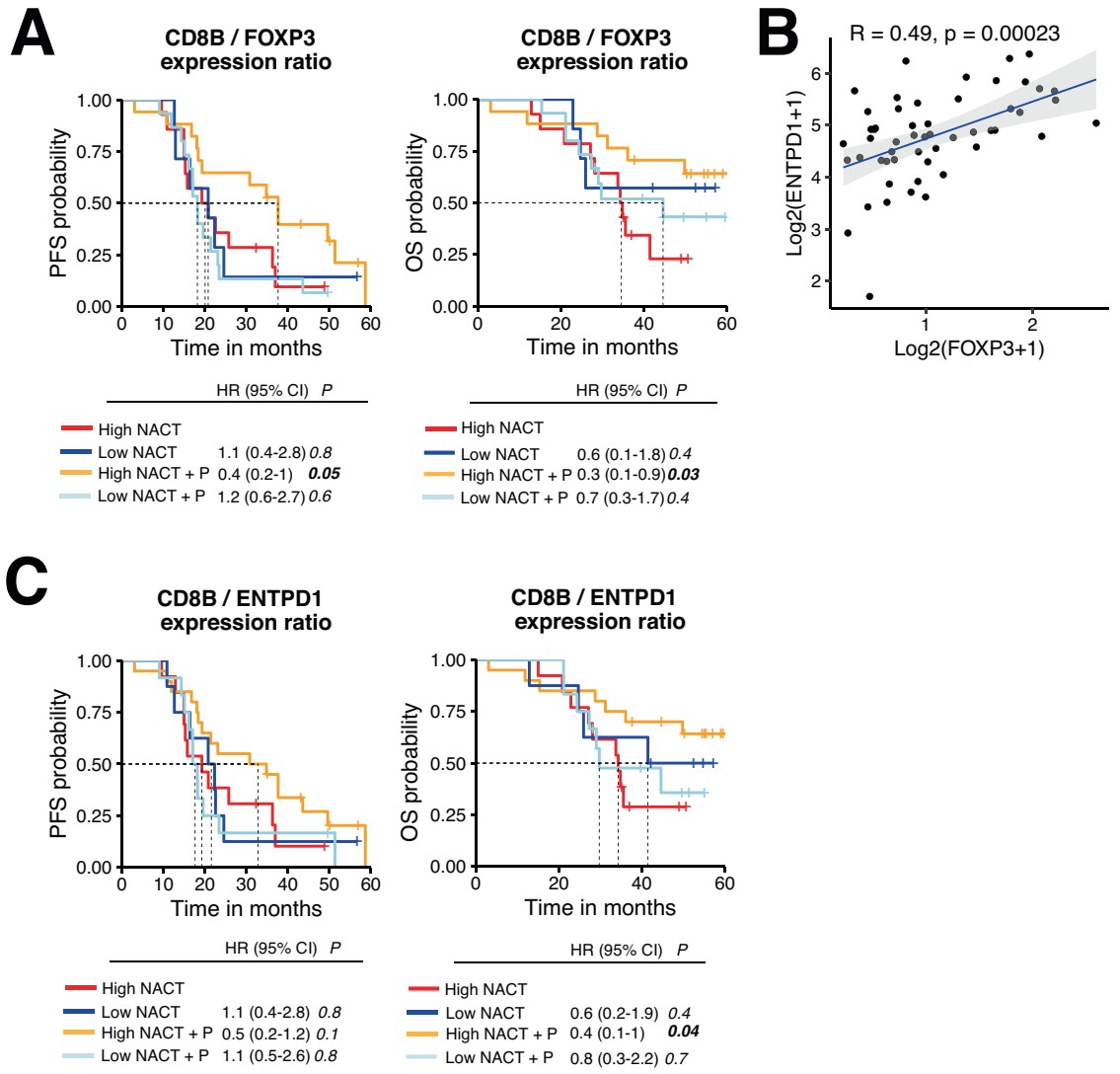

**Fig. 5 | A high *CD8B/FOXP3* gene expression ratio is associated with increased survival in patients treated with NACT + P. A, C** PFS (left) and OS (right) curves according to the *CD8B/FOXP3* expression ratio (High NACT (red), $n = 14$; Low NACT (dark blue), $n = 7$; High NACT + P (orange), $n = 17$; Low NACT + P (light blue), $n = 15$) (**A**) and the *CD8B/ENTPD1* expression ratio (High NACT (red), $n = 13$; Low NACT (dark blue), $n = 8$; High NACT + P (orange), $n = 20$; Low NACT + P (light blue), $n = 12$)

(**C**) in each arm. Patients were stratified based on the best cutoff. Statistical comparison of survival curves for NACT + P High vs. NACT High was performed using the likelihood ratio test. **B** Spearman correlation between log2(TPM + 1) FOXP3 and ENTPD1 gene expression. Error bands represents the 95% CI as a shaded gray area. Source data are provided as a Source Data file.

## Discussion

Although HGSC is associated with a strong immunogenicity[2,34], PD-1/PD-L1 blockade has so far had little effect on patient survival in either frontline or relapse settings[9–11]. A better comprehension of the biological effects of PD-1 blockade on the tumor microenvironment of HGSC and the identification of factors promoting response or resistance to treatment is therefore essential to guide future developments of precision immunotherapy in this deadly disease. Here, we report a comprehensive study combining transcriptomic and in situ multiplexed immunofluorescence data on paired tumor samples collected prospectively during a randomized clinical trial of HGSC patients, receiving either the standard therapy (NACT) or NACT + Pembrolizumab ± Bevacizumab in the first line setting. Using this multiomics approach, we confirmed that chemotherapy remodel the tumor microenvironment by increasing tumor infiltration by lymphocytes and showed that the addition of Pembrolizumab amplified these effects with signs of T cells reinvigoration and increased CD8+PD-1+ T cells in the vicinity of tumor cells. By selecting robust independent variables of response to treatment using the LASSO method, we found

that a high monocyte signature score, a low endothelial signature score and a high *CD8B/FOXP3* expression ratio were associated with better survival. The combination of these biomarkers was a better predictor of response than the previously reported biomarkers PD-L1 and intra-epithelial CD8+ T cells. Finally, we provided evidence that VEGFR2 and CD39 (*ENTPD1*) are involved in resistance to Pembrolizumab.

Expression of genes related to cytotoxic CD8+ T cells and pathways associated with T cell differentiation and function were more frequently increased following NACT + P than following NACT alone, revealing that Pembrolizumab has a specific impact of on the T cell infiltrate and allows its reinvigoration. This was confirmed by spatial analysis of the tumor microenvironment by multi-IF revealing a significant increase in PD-1-expressing CD8+ T cells in the vicinity of tumor cells following NACT + P but not chemotherapy alone. CD8+PD-1+cells have been demonstrated in various cancers to contain tumor-reactive T cells[25,35–39], and, to our knowledge, have not been described to be increased following NACT alone[6,7,40], indicating that the increase in CD8+PD-1+ T cells is essentially promoted by PD-1 blockade. This is

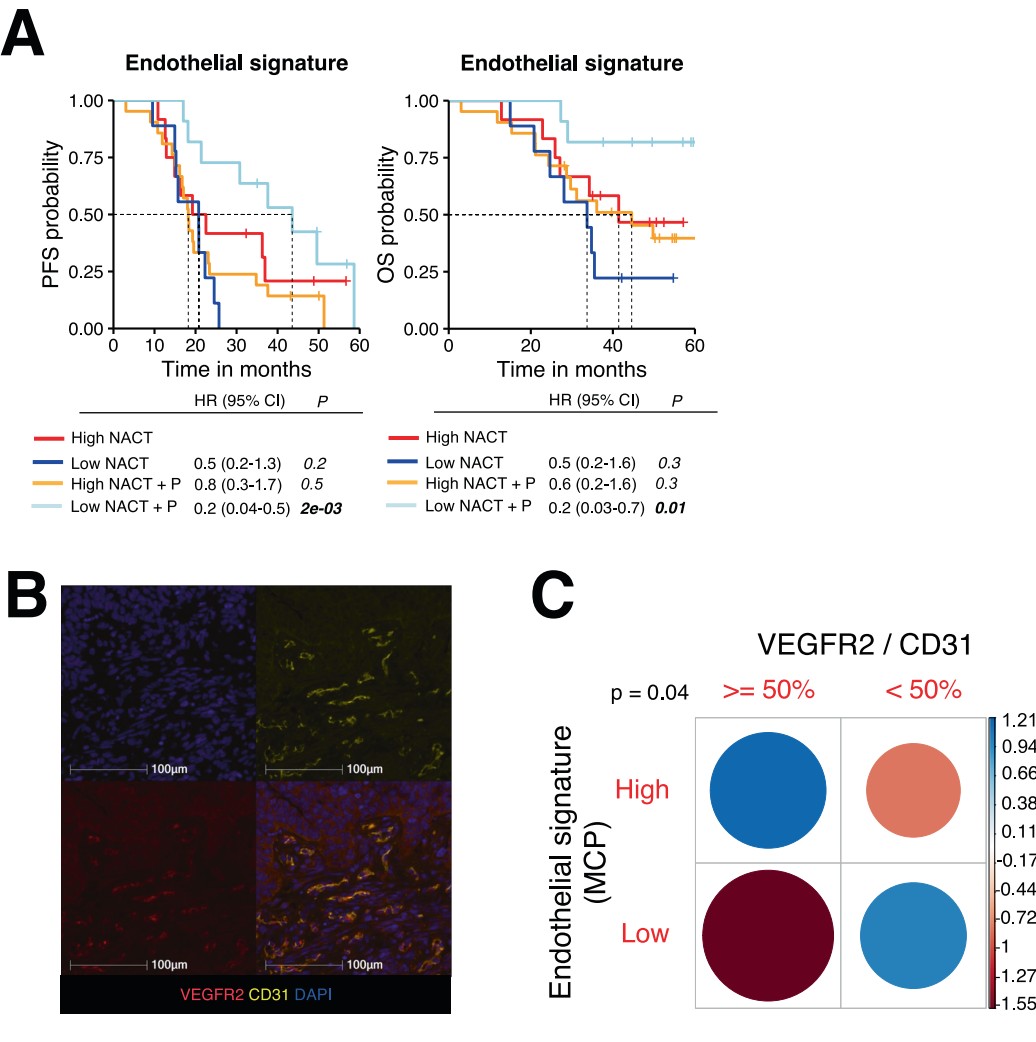

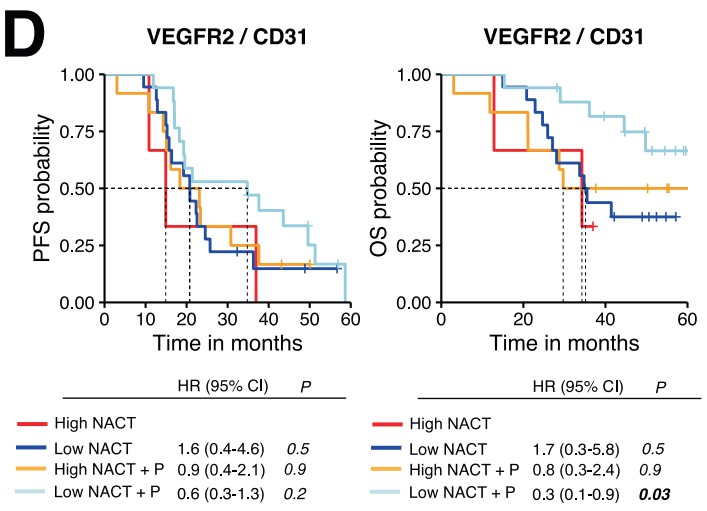

consistent with results in pancreatic cancer showing that Nivolumab increased intra-tumoral CD8$^+$PD-1$^+$ T cells in post-treatment samples[41]. We also showed that a high density of CD8$^+$PD-1$^+$ T cells after NACT + P tended to be associated with OS, which confirms previously reported data by Thommen and colleagues in a small cohort of non-small cell lung cancer patients treated with PD-1 blockade[42]. We also observed higher *CD8B/FOXP3* gene expression ratios in non-progressor patients

under the combination NACT + P compared to progressors, suggesting that a high ratio of CD8$^+$ effector T cells to CD4$^+$ regulatory T cells is associated with response to treatment. Even though the presence of regulatory T cells has already been reported to be associated with resistance to ICI[43], strategies aiming at depleting regulatory T cells using anti-CTLA-4 antibodies have not been successful in HGSC[44,45]. Interestingly, in our study, NACT + P was more effective than NACT

**Fig. 6 | High expression of *KDR*/VEGFR2 is associated with resistance to NACT + P. A** PFS (left) and OS (right) curves according to the expression of the endothelial gene signature in each arm (High NACT (red), *n* = 12; Low NACT (dark blue), *n* = 9; High NACT + P (orange), *n* = 21; Low NACT + P (light blue), *n* = 11). Patients were stratified based on the best cutoff. Statistical comparison of survival curves for NACT + P Low *vs.* NACT Low was performed using the likelihood ratio test. **B** Representative image of a CD31/VEGFR2 multiplex IF staining showing expression of VEGFR2 (red) on endothelial cells (CD31 in yellow). **C** Correlation between the endothelial signature expression score and VEGFR2 expression on CD31[+] cells assessed by multi-IF. Color intensity and the size of the circle are

proportional to the correlation coefficients. Positive values are in blue and denote a positive association while negative values are in red and denote a negative association. Pearson's $\chi^2$ test *p*-value is displayed. No adjustment for multiple comparisons was made. **D** PFS (left) and OS (right) curves according to the expression of VEGFR2 on CD31+ endothelial cells ( ≥ 50% vs <50%) in each arm (High NACT (red), *n* = 3; Low NACT (dark blue), *n* = 18; High NACT + P (orange), *n* = 12; Low NACT + P (light blue), *n* = 17). Statistical comparison of survival curves for NACT + P Low *vs.* NACT Low was performed using the likelihood ratio test. Source data are provided as a Source Data file.

alone in patients with a high *CD8B/ENTPD1* (CD39) ratio both in terms of PFS (trend) and OS, indicating that CD39 could be involved in resistance to Pembrolizumab. This observation may seem counter-intuitive, as CD39 is a marker of tumor-reactive exhausted CD8[+] T cells[46,47] and has been associated with response to ICI in lung cancer[48]. Yet, CD39 is also expressed at high levels by Tregs and a subset of regulatory T cells co-expressing PD-1 and CD39 has been associated with increased suppressive capacity in HGSC[49,50]. Our own FCM data revealed that CD4[+] Tregs infiltrating HGSC express far higher levels of surface CD39 than CD4[+] or CD8[+] effector T cells. These observations favor the hypothesis that the *CD8B/ENTPD1* expression ratio may be linked to tumor infiltration by CD8 effector T cells *versus* activated (CD39[high]PD-1[+]) Tregs. This could explain the absence of clinical benefit of adding Pembrolizumab to NACT in the NeoPembrOv trial and pave the way for future promising combinations targeting Tregs, such as the use of anti-CD39 antibodies. Recently, TLS have been reported to be key determinants of the response to ICI in various cancers[51–53]. Intriguingly, we did not find any association between response to NACT + P and the presence or density of TLS in tumors collected either before treatment or at debulking surgery. Yet, in the present study, Pembrolizumab was evaluated in a neoadjuvant setting in combination with chemotherapy. We may thus hypothesize that corticosteroids given systematically at a relatively high doses during NACT in HGSC might impair their prognostic value, as shown in lung cancer[54].

In addition, we showed that a high expression of VEGFR2 on endothelial cells was associated with resistance to immunotherapy but not to Bevacizumab. Endothelial cells have been described to inhibit tumor immunity by downregulating molecules required for T-cell extravasation, by upregulating inhibitory receptors on T cells and by inducing T-cell apoptosis through TRAIL and FasL[55,56]. VEGFR2 blockade and subsequent normalization of the tumor endothelial barrier was reported to enhance T cell infiltration and function[57–60]. In HGSC, the antiangiogenic therapy consists of an anti-VEGFA antibody (*i.e.*, Bevacizumab). In our study, however, Bevacizumab (used in almost 90% of patients) did not enhance the effect of NACT + Pembrolizumab compared to NACT alone, suggesting that blocking VEGFR could be more effective than blocking VEGFA as an antiangiogenic. This is supported by different data from the literature. First, in malignancies with high VEGF levels such as HGSC[61], T-cell development is inhibited by VEGFR2, further highlighting that VEGFR2 could be an interesting target. Second, both PDGF and VEGF are associated with poor prognosis in HGSC[62,63]. As both PDGF and VEGF bind to VEGFR2[33], the use of a VEGFR2 inhibitor could be more beneficial than a VEGFA inhibitor as it would also inhibit the effect of PDGF on progression. Finally, the use of ramucirumab, a VEGFR2 inhibitor, was reported to increase response rates to Pembrolizumab in lung cancer patients that had progressed after chemotherapy + ICI[64]. Interestingly, TAMs are an important source of pro-angiogenic factors, such as VEGF and PDGF[65,66], and some TAMs also express VEGFR2[67], indicating that PDGF could drive their recruitment or their function[68]. As VEGFR2 blockade has been shown to reduce macrophage infiltration into tumors[67], using VEGFR2 inhibitors in HGSC could be appealing as it may impact both angiogenesis and type 2 macrophage infiltration/function.

Indeed, we showed that the monocyte and type 2 macrophage signatures were negatively correlated and were inversely associated with response and prognosis. While these two immune populations share a common origin, *TREM2* is highly expressed in type 2 macrophages compared to monocytes[69], especially in HGSC compared to other solid tumors[70,71]. In our study, a high *TREM2* expression was correlated with poorer survival. It has been shown that TREM2[+] macrophages are correlated with an exhausted T cell state in the tumor microenvironment[71] and that an anti-TREM2 antibody has the ability to deplete macrophages and increase T cell function in the tumor infiltrate[72]. This suggests that targeting TREM2[+] macrophages could be an interesting therapeutic avenue in HGSC, a possibility that is currently being investigated in a clinical trial (ClinicalTrials.gov Identifier: NCT04691375).

Some limitations of the present study include the relatively small number of patients included in the phase II trial (*n* = 91), insufficient quality of certain tumor samples for translational analyses and the absence of samples for some patients, either because they were not eligible for IDA (*n* = 4) or tumor resection was not possible (CC-3, *n* = 19). Nevertheless, this study highlights potential biomarkers and generates several hypotheses that warrant to be confirmed and further explored in an external cohort of HGSC patients treated with ICI and with public transcriptomic data. We revealed that Pembrolizumab remodels the immune TME, supporting that it should not be discarded in HGSC despite its low benefit so far. Instead, we propose some ways to reprogram the immune and non-immune TME of HGSC to increase responsiveness to existing ICI and envisage combination therapies. In particular, potentiating effector T cells by depleting/neutralizing regulatory T cells expressing CD39 and/or targeting VEGFR2[+] endothelial cells could be options for sensitizing HGSC to ICI.

## Methods
### Patient cohort
The NeoPembrOv trial (NCT03275506) is an open-label, randomized non-comparative phase II trial for which primary outcomes are published[22]. Eligible patients had newly diagnosed (by laparoscopy or laparotomy) histologically confirmed FIGO Stage IIIC or IV epithelial ovarian, fallopian tube, or primary peritoneal carcinoma that was high-grade serous or endometrioid. Patients had to be considered unsuitable for primary debulking surgery and planned for NACT followed by cytoreductive interval debulking surgery (IDS) aiming for no residual disease. All patients provided written informed consent before undergoing any study-specific procedures. The study was performed in accordance with the ethical principles of the Declaration of Helsinki, the International Conference on Harmonization/Good Clinical Practice guidelines, and the Public Health Code in France. This French trial received a favorable opinion from a French national ethics committee Comité de Protection des Personnes (CPP) Nord Ouest II based in Amiens. Eligible patients were randomized in a 1:2 ratio to receive chemotherapy alone (Carboplatin + Paclitaxel, control arm) or in combination with Pembrolizumab (investigational arm). Bevacizumab was given at the investigator's discretion after IDS and was prescribed after surgery in almost 90% of patients. Tissue from a newly obtained (<8 weeks before starting study treatment) core or excisional biopsy of a tumor lesion, defined as pre-treatment tumor sample, was collected

for all patients. Tumors were then sampled at IDS and were defined as post-treatment tumor samples. With the help of the GINECO, a cooperative intergroup specialized in clinical and translational research in the field of gynecological oncology, 88 of the 91 patients included had a FFPE tumor sample available at baseline (96.7%) and 68/91 at IDS (74.7%). Among the sixty-eight patients with paired samples before and after NACT (4 cycles), 64 had sufficient material for multi-IF staining (4 patients were dismissed due to poor quality tissue—ascites, and necrosis). Of the 68 patients, 57 had sufficient amounts of tumor sample to perform both IF and molecular analyses, and underwent RNAseq. Fifty-three patients had paired samples available for analysis (no library preparation failure).

PD-L1 expression was determined in the baseline pre-treatment tumor tissue samples collected during diagnostic laparoscopy. PD-L1 IHC diagnostic assay was performed on each specimen according to the manufacturer's instructions: Ventana SP263 (rabbit monoclonal primary anti–PD-L1 antibody, prediluted, Ventana Medical Systems, Tucson, AZ) on the Benchmark XT staining systems and Ultra with OptiView Universal DAB Detection Kit (Ventana Medical Systems) by PATHEC, our pathology platform. Interpretation of PD-L1 expression was performed by a trained pathologist specialized in gynecology tumor analysis. PD-L1 expression in the tumor cell membrane and membrane and/or cytoplasm of tumor-associated mononuclear inflammatory cells such as lymphocytes and macrophages was scored. The CPS (combined positive score) was defined as the total number of tumor cells and immune cells (including lymphocytes and macrophages) stained with PD-L1 divided by the number of all viable tumor cells, then multiplied by 100. Samples with <10% tumor and immune cell staining were PD-L1 CPS-negative and samples with ≥10% tumor and immune cell staining were considered to be PD-L1 CPS-positive.

At the data cutoff for the updated analysis (15 June 2023), the median duration of follow-up was 52.4 months (range 24.6–62.7 months). PFS was similar in the two treatment arms (median 20.8 months [95% CI 15.0–25.7 months] in the control arm and 19.4 months [95% CI 17.0–26.7 months] in the investigational arm; Fig. 1A). OS results were immature.

## Multiplex immunofluorescence tissue staining

A hematoxylin and Eosin (H&E) stained slide of each sample was examined by a trained pathologist to confirm tissue quality, select samples for multi-IF staining and annotate the tumor area. Seven-color sequential multi-IF staining was performed with the BOND RX stainer (Leica Microsystems, Buffalo Grove, Illinois) using previously validated Ab panels and a control (tonsil) section was included in each staining batch to inspect the overall staining fidelity for all markers. Two seven-color panels were conducted by the Research Pathology Platform East (N. Gadot), one focusing on B cells, TLS and ASC and another one on T cell populations and activation status (Supplementary Table 5). After deparaffinization, rehydration and antigen retrieval, 4 μm FFPE tumor sections were sequentially stained with each primary antibody, followed by OPAL-HRP secondary antibody incubation, and then revealed with tyramide signal amplification and OPAL fluorophores (Akoya Biosciences) and the same cycle was reproduced until staining with the last Ab of the panel. A manual five-color multi-IF staining was also performed to study VEGFR2 expression. Slides were counterstained with spectral DAPI (Akoya Biosciences) and cover slips were mounted using Prolong Gold medium (Invitrogen, Paisley, UK). Whole slides were imaged at a 20x magnification using the Vectra Polaris multispectral scanner (Akoya Biosciences) and at a 40x magnification for illustration purposes.

## Analysis of multi-IF digital images

The panel focusing on TLS and ASC was quantified in collaboration with Keen Eye Technologies (Paris, France) using deep learning image analysis algorithms and whole slide raw images. After manual annotation by a trained biopathologist of the areas to be analyzed, tissues were segmented in tumor, stroma, no tissue area (including fat tissue) and artifacts (necrosis, tissue folding, massive blood cell areas, non-specific auto-fluorescent structures). Specific algorithms were designed and optimized to automatically detect and segment B follicles and TLS to quantify their number and surface. TLS were defined as dense aggregates of CD20+ cells (B-follicle) adjacent to areas rich in T cells (CD3 + ) and dendritic cells (DC-LAMP + ). Cells were segmented based on DAPI nuclear staining. To specifically increase the robustness of cell segmentation of tumor cells, the CK-Opal480 channel was also considered. Finally, deep-learning phenotyping algorithms were designed and optimized for each marker. Cell density as well as cell proximity metrics (nearest neighbors and count within 20 μm) of each whole slide were generated from those analyses.

For the T and VEGFR2 panels, digital images were visualized with the Phenochart viewer (Akoya Biosciences) and representative ROIs (median 6, range [1–10] depending on the tumor surface) were selected by two observers. After spectral unmixing using the synthetic library of the inForm software (Akoya Biosciences), tissue and cell segmentation was performed as for the other multi-IF panels to classify tumors, followed by cell phenotyping. A machine-learning algorithm was trained by user-specified tissue annotations aided by the signal from the epithelial marker to accurately segment tumor tissue versus stromal tissue and background; using the endothelial cell marker to segment CD31+ area versus CD31- area for the T and VEGFR2 panel, respectively, as well as individual cells using the nuclear DAPI signal. The algorithm was trained on ROIs from 11 slides representative of the cohort and validated on an additional set of 22 slides before launching the batch analysis of all slides. After manual inspection, the algorithm was again optimized, if necessary, until correct segmentation and phenotyping (>80% of cells correctly identified). For the VEGFR2 panel, cells with a confidence <75% were removed from the analysis. Finally, the phenotyping data were exported from inForm and tabulated reports including cell densities, marker intensity, and spatial analyses (nearest neighbor, count within a 20 μm radius) were obtained using the R package phenoptr or PhenoptrReports (Akoya Biosciences).

## Flow cytometry

Multiparametric FCM analyses were performed on single-cell suspensions derived from ovarian tumors. The FCM panel used to assess T-cell differentiation relied on the use of anti-human antibodies against CD3, CD4, CD8, CD45RA, FOXP3, CD39, and a viability marker (Supplementary Table 6). After gating on CD45+ and CD3+ T cells within viable cells, CD4 Teff were defined as CD4+/FOXP3- cells. CD4 T reg were defined as CD4+/FoxP3+ cells and T CD8 as CD8+ cells (Supplementary Fig. 7).

## RNA sequencing

**Library preparation and sequencing.** Biological samples were prepared by BB-0033-00050, CRB Centre Léon Bérard, Lyon France. Total RNA from FFPE tissues was extracted using 2 to 5 10-μm FFPE sections from pre- and post-treatment samples with FormaPure RNA kit (Beckman Coulter). RNA concentration and absorbance were analyzed using a NanoDrop spectrophotometer (Thermo Fisher Scientific), and quality profiles assessed on Tapestation (Agilent Technologies). Libraries were prepared with the TruSeq RNA Exome kit (Illumina) and sequenced on the NovaSeq sequencing platform (Illumina) of the CRCL with a paired-end protocol (76 bp) and a target coverage of 64 M bp reads by the cancer genomics platform. The primary base call files were converted into FASTQ sequence files using the bcl2fastq converter tool.

## Transcriptomic data analysis

**Sequence alignment and normalization.** Raw sequencing reads were aligned on the human genome (GRCh38) with STAR[73] (v2.7.3a). Gene expression was quantified using Salmon[74] (v1.0.0) and the annotation

of known genes from gencode v29. Non-expressed and lowly expressed genes (<10 counts) were removed. We calculated the TPM (Transcripts Per Million) by normalizing each gene length by the total read counts.

**Quality control.** RNA-seq quality was evaluated by considering the total number of reads, alignment rate and duplication rate. Five samples did reach qiality control standards and were removed.

**Differential analysis.** All analyses were performed in R programming language (v4.0.2). DESeq2 (v1.28.1) was used to identify differentially expressed genes (DEGs) between post- and pre-treatment samples for each treatment arm separately[75]. As already published[76,77], only genes with a fold-change > 1.5 and a Benjamini and Hochberg adjusted $p$-value for multiple testing <0.05 were considered. Functional analyses (GO BP, GO MF, KEGG, MsigDB C2, and Hallmark collection) were performed using ClusterProfiler (v3.16.1). Only significant pathways ($_{adj}P$ <= 0.05) were considered for graphical representation using ggplot2 (v3.3.2).

**Immune profiling.** Tumor immune infiltration was inferred from gene signature scores using multiple computational tools, including MCP-counter[23] and quanTIseq[24]. Indeed, to estimate immune cell contents of tumor samples two types of methods exist, namely marker gene-based (such as MCP-counter) and deconvolution-based (such as quanTIseq) approaches. On the one hand, marker gene-based approaches use a list of genes that are characteristic of a cell type. Deconvolution methods, on the other hand, uses a system of equations that describe the gene expression of a sample as the weighted sum of the expression profiles of the admixed cell types generating absolute scores that can be interpreted as cell fractions and be compared inter- and intra-sample[78]. For MCP-counter, unique gene expression and specific genes were generated: *PTPRC* expression as a surrogated of immune cells; *EPCAM* expression as a surrogate of tumor cells; the sum of *IGHA1* and *IGHA2* as a marker of IgA isotype expression; the sum of *IGHG1, IGHG2, IGHG3,* and *IGHG4* as a marker of IgG isotype expression; *CXCL13* a surrogate of TLS[79]; *JCHAIN* as a specific marker of plasma cells[80]; and the sum of *TNFRSF17* (marker of long-lived plasma cells[81]) + *MZB1* (key effector of the Blimp1 regulatory network in plasma cells[82]) to identify plasma cells. To infer the ratio of CD8 + T cells to Treg, we calculated the *CD8B/FOXP3* expression ratio and privileged *CD8B* over *CD8A* as this gene is more specific of CD8 + T cells and is included in the CD8 + T lymphocytes signature of MCP counter[83,84].

**Heatmap.** ComplexHeatmap (v2.4.3) was used to represent the differences between post-and pre-treatment immune infiltrates.

**ssGSEA.** ssGSEA was used to score gene expression programs of neoantigen-specific tumor-infiltrating lymphocytes with the list of genes previously reported[26] by computing the difference between a weighted Empirical Cumulative Distribution Function (ECDF) of the genes in the signature and the ECDF of the remaining genes.

## Statistics and reproducibility
**Least absolute shrinkage and selection operator.** Variables associated with response were integrated into a LASSO method using glmnet R package (v4.0-2) for more robust predictive signatures. This type of model was favored over a multiple logistic regression model as the objective was to select useful rather than redundant features, in a context of numerous predictor variables with few events, and multicollinearity[85]. All the variables were categorized with known thresholds for clinical factors and with the optimal cut-off for exploratory variables. To limit overinterpretation and overfitting, 100 bootstrap samples were generated and only variables selected through all the 100 corresponding lasso Cox models were retained in the final

model. After selection of robust independent variables with the LASSO model, a generalized linear model was performed to estimate the effect on the outcome of the independent variable (β) and the standard error and the associated $p$-value. Areas under the curve (AUC) for final variable selection were compared to evaluate model robustness using ROCR (v1.0-11) and pROC (v1.16.2).

**Univariate Cox model.** Cox univariate models were run using the coxphf function (R package v1.13.4) to identify prognostic and predictive biomarkers of response for each treatment arm. Optimal cut-offs for continuous exploratory variables scores were chosen based on a maximally selected rank statistics using survminer (v0.4.9). Comparison of survival curves was performed using the likelihood ratio test due to sample size.

Statistical analyses were performed using R programming language (v4.0.2). Two-sided Wilcoxon signed-rank test was used for paired-sample comparisons, while two-sided Wilcoxon rank-sum test was used to compare independent groups. Due to the exploratory nature of the study, $_{unadj}p$-values < 0.05 were considered significant, with stars corresponding to *$p < 0.05$; **$p < 0.01$; ***$p < 0.001$ and ****$p < 0.0001$. If no stars are indicated, no statistically significant difference was found. False discovery rates (FDR) were added.

### Reporting summary
Further information on research design is available in the Nature Portfolio Reporting Summary linked to this article.

## Data availability
Processed RNA sequencing data presented in the manuscript has been deposited in the Gene Expression Omnibus (GEO) database under the following GEO ID: GSE227666. Processed data corresponds to reads aligned to the reference genome to obtain the gene expression matrix. To ensure samples anonymisation, different IDs are used between expression matrix and patients information available in supplementary data. Data sharing in a public repository was not planned at the start of the study. Per European and French regulations for personal data privacy, this is not permitted without having informed the study participants which was not done. This is also linked to a confidentiality agreement with MSD, who provided the drug and funding. This agreement aims to guarantee protection for the company about potential sub-licensable or patentable information/discovery. Requests to access the deidentified data for further scientific use can be sent to ARCAGY-GINECO (Sébastien Armanet sarmanet@arcagy.org) will be considered on a case-by-case basis in a timely manner beginning 3 months and ending 5 years after this article publication. The request must contain a proposal with scientific and methodologically justified objectives. A Data Transfer Agreement will be established to provide a formal framework regarding the use of the data. The remaining data are available within the article source data file. Source data are provided with this paper.

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

## Acknowledgements

We thank the patients and their families for their participation in this study, as well as the study teams at each of the study sites. Translational research was supported by the ARC Foundation—DefImmunPD-1 project [PGA1 RC20170205307]. The clinical trial was supported by MSD. The funding agencies/companies have not participated in the study design or analysis of the results. We also thank Déborah Cardoso, Alexandre Degnieau, Christine Montoto-Grillot from ARCAGY GINECO, the sponsor, for clinical study design, clinical data collection and analysis and for logistic help for translational research. We thank the Biological Resources Center of the Léon Bérard Cancer Center, Laëtitia Odeyer and Amélie Vermorel for performing the PD-L1 staining and Brigitte Manship for diligent proof-reading of the manuscript. We are also grateful for the technical support from the core facilities at CRCL, in particular the Research Pathology Platform East (N. Gadot) and Cancer Genomics Platforms, the LYriCAN+ (Grant INCa-DGOS-INSERM-ITMO cancer_18003) for funding the PD-L1 analysis and the Rhône Alpes Auvergne region (Grant IRICE RRA18 010792 01 – 10365) for initial funding of the LICL multi-IF platform. This work was performed within the framework of the LABEX DEVweCAN (ANR-10-LABX-0061) of the University of Lyon and beneficiated from help from the Institut Convergence Plascan (ANR-17-CONV-0002).

## Author contributions

O.L.S., C.C., I.R.-C. and B.D. conceived and designed the translational research, coordinated procedures, analyzed, and interpreted the data, and wrote the manuscript. A.M.S., M.-A.M.-R., E.C., O.D. and I.R.-C. included and treated patients in the trial. P.-A.J., I.T., and G.B. reviewed all pathological cases. I.T. analyzed the PD-L1 expression. M.A. and O.L.S analyzed and interpreted RNA-sequencing data and wrote the manuscript and prepared figures and tables. J.B., S.B., Y.L., M.B., L.J., S.B. and O.L.S. analyzed and interpreted IF data. J.C. and C.M.-C. analyzed FCM data. E.T., M.T. and I.G.-F. supervised angiogenic analyses. M.-H.S. and I.L-G. participated in the translational data interpretation. A.F. and E.T. supervised the bioinformatics analyses. S.T.-E. coordinated tissue biobanking. I.R.-C. and B.D. jointly supervised the work. All authors reviewed the results, edited, and approved the final version of the manuscript.

## Competing interests

O.L.S. reports honoraria from MSD and Clovis, and travel/accommodation/expenses from Eisai. E.C. reports the following, all for an immediate family member: employment with Sanofi; consulting/advisory role for MSD, BMS, Ipsen, and AstraZeneca; speakers' bureau for BMS, Ipsen, AstraZeneca, and Janssen; research funding from Pfizer; and travel/accommodation/expenses from Janssen. I.R-C reports personal honoraria from Agenus, Blueprint, BMS, PharmaMar, Genmab, Pfizer, AstraZeneca, Roche, GSK, MSD, Deciphera, Mersana, Merck Sereno, Novartis, Amgen, MacroGenics, Tesaro, and Clovis; honoraria to her institution from GSK, MSD, Roche, and BMS; consulting/advisory roles for AbbVie, Agenus, Advaxis, BMS, PharmaMar, Genmab, Pfizer, AstraZeneca, Roche/Genentech, GSK, MSD, Deciphera, Mersana, Merck Sereno, Novartis, Amgen, Tesaro and Clovis; research grant/funding from MSD, Roche and BMS (self) and MSD, Roche, BMS, Novartis, AstraZeneca and Merck Sereno (to institution); and travel support from Roche, AstraZeneca, and GSK. Other authors declare no competing interests.

## Additional information

[1]"Cancer Immune Surveillance and Therapeutic Targeting" Laboratory, Cancer Research Center of Lyon, INSERM 1052—CNRS 5286, Centre Léon Bérard, Université de Lyon, Université Claude Bernard Lyon 1, 69008 Lyon, France. [2]Lyon University, Université Claude Bernard Lyon 1, Centre Léon Bérard, 69008 Lyon, France. [3]National Investigators Group for Ovarian and Breast Cancer Studies, Paris, France. [4]Present address: Department of Medical Oncology, Centre Léon Bérard, 69008 Lyon, France. [5]Present address: Lyon Immunotherapy for Cancer Laboratory (LICL), Cancer Research Center of Lyon, Centre Léon Bérard, 69008 Lyon, France. [6]Present address: Université Paris Cité, Inserm, PARCC, F-75015 Paris, France. [7]Present address: Department of Anatomopathology, Centre Léon Bérard, 69008 Lyon, France. [8]Present address: Department of Anatomopathology, AP-HM, Marseille, France. [9]Present address: Department of Anatomopathology, University hospital of Toulouse, Toulouse, France. [10]Present address: Department of Medical Oncology, Institut Jean Godinot, Reims, France. [11]Present address: Department of Medical Oncology, Centre Jean Perrin, Clermont-Ferrand, France. [12]Present address: Department of Medical Oncology, Centre François Baclesse, Caen, France. [13]Present address: Department of Medical Oncology, Hôpital Privé Jean Mermoz, Lyon, France. [14]Present address: Keen Eye Technologies—Paris, France, now Tribun Health, Paris, France. [15]Present address: Department of Oncology, Hôpitaux universitaires de Genève, Faculty of Medecine, Center of Translational Research in Onco-Hematology, Swiss Cancer Center Leman, Geneva, Switzerland. [16]Present address: Human Biological Resources, Centre Léon Bérard, 69008 Lyon, France. [17]Present address: Synergie Lyon Cancer, Gilles Thomas Bioinformatics Platform, Centre Léon Bérard, CEDEX 08, F-69373 Lyon, France. [18]Present address: Inserm U830, DNA Repair and Uveal Melanoma (D.R.U.M.) Team, Institut Curie, PSL Research University, 75005 Paris, France. [19]These authors contributed equally: Bertrand Dubois, Isabelle Ray-Coquard. ✉e-mail: bertrand.dubois@lyon.unicancer.fr; isabelle.ray-coquard@lyon.unicancer.fr

