## [Peer Review File · Nature Communications]

Reviewers' Comments:

Reviewer #1:

Remarks to the Author:

Thank you for the opportunity to review this interesting work looking at tumor-based biomarkers of benefit to carboplatin/paclitaxel with or without pembrolizumab in patients with newly diagnosed HGSOC. My comments are as below:

1) Please discuss the limitations of the study in the discussion section, including the fact that only 64 of 91 patients had sufficient paired tissue for analysis mIF and only 53 for RNAseq. This may have introduced bias into your results. In the supplemental, it would be helpful to have venn diagram of how many samples were obtained from each arm of the study.

2) For R/NR, how was the 24 month PFS mark selected as clinical endpoint? In the clinical paper, the median duration of follow up was 22 months.

3)It's also important to note that the use of bevacizumab may also impact your clinical endpoint as bevacizumab has been shown to prolong PFS (although not OS). It would be helpful for the authors to include what percentage of R/NR got bevacizumab, and is it controlled across the two arms? Was bev use incorporated in the statistical models for the predictive biomarkers?

4) Why was CK selected as your marker for tumor cells for mIF as opposed to PAX8?

5) In the discussion, the authors should acknowledge that regulatory T cells, as delineated by FOXP3 positivity, have previously been shown to be a negative predictive biomarker for IO response and this is not a novel finding. They might also discuss some of the clinical data for t reg depletion strategies, although they have not been successful thus far.

6) It would be helpful to understand the data for the CD8/CD39 predictive ratios a little bit better, as this is somewhat counterintuitive. Prior data have demonstrated that Tex are the cells that are reinvigorated by PD-1/L1 blockade, and prior work suggests that CD39 is a marker of T cell exhaustion (PMID: 29066514). Were other T cell exhaustion markers upregulated in the RNA seq data in the patients who were responders? or regulatory T cell markers? Overall, this is an area that needs some attention and fleshing out to understand potential mechanisms.

7) The nature of the color coding in Figure 1 panel C is unclear to me - why are some of the bars red, blue and purple?

8)Please use asterisks across all the figures to denote statistical significance.

9) Figure 2 Panel B and E don't add much to the scientific story and could probably be removed for brevity and clarity.

10) In figure 5, the PFS and OS lines are all overlying because the figure is so small; it would improve legibility if these are made bigger.

Reviewer #2:

Remarks to the Author:

The investigation by Le Saux and colleagues used paired samples from the NeoPembrOv randomized phase II trial for RNA-seq and multiplexed immunofluorescence to explore the impact of neoadjuvant chemotherapy (NACT) ± Pembrolizumab (P) on the tumor environment and identify biomarkers for response.

The strength of this study is the use of biospecimens from the above noted randomized clinical trial. Unfortunately, some of the methods used make interpretation of the results difficult. Please see below.

Critiques

1. It is difficult to evaluate the results in their current form due to the inappropriately low

statistical significance cutoff ($p < 0.05$) used for the primary RNAseq analysis causing significant likelihood of false positive results due to multiple comparisons. The pathway enrichment analysis (Fig 1C) does appear to be statistically significant (with adjusted p values), however this analysis is based on the primary DEG from the RNAseq.

2. How was tumor percentage controlled for in the RNA seq analyses between pre and post NACT samples? It would be expected that the post samples may have lower tumor percentage which could impact DEG results.

3. The authors should discuss how interaction between their pre-treatment biomarkers (figure 3C). It seems that CD8B/FOXP3 ratio and endothelial cell signature showed a significant interaction with treatment, suggesting that they were predictive biomarkers of response, while the monocyte signature did not. However, the prediction model used all three. It would be helpful to explain the rationale.

4. Manuscript would benefit from scientific editorial review to enhance comprehensibility (e.g. "To precise the predictive impact of the endothelial cell gene signature variable, we categorized patients into "high" and "low" expressors using the optimal threshold, which was close to the median, with 62.3% vs 37.7% of patients having a high or low gene signature expression score, respectively.")

5. The authors should temper the language in the discussion to reflect the correlative nature of their observations instead of implying causation. For example, in the Discussion they make the statement "Using this multi-omics approach, we found that Pembrolizumab induced the recruitment of intra-epithelial CD8+PD-1+ T cells." In fact, their data does not support this assertion. There are multiple other examples in this type of over-stating conclusions in the Discussion section.

Reviewer #3:

Remarks to the Author:

A remark diagram should be provided to display how many patients were randomized to each treatment arm and then how many patients had sufficient tumor for determination of immune cell populations in each arm.

Results Line 97 State how many of the 64 patients were assigned to NACT and how many were assigned to NACT + PEM.

Results Line 107 and Figure 1 B, C: It is unclear why p-value < 0.25 was chosen as a cutoff point for declaring significance. Sixteen comparisons are being reported in Figure 1B and 12 comparisons are being reported in Supplementary Figure 1C. Instead of loosening the threshold for declaring significance, the p-value should be adjusted to control the false positive rate. Thus, none of the comparisons discussed in lines 94-119 are significant. Any percentages reported such as the percentage of patients with increased expression of the CD8+ T cell gene signature post-treatment should be accompanied by a 95% confidence interval so that the reader can see the variability in that estimate.

Results Line 136-166. It is not clear how changes in density pre and post treatment were analyzed. Was the percent change, fold change or difference examined? What hypothesis test was used to assess changes. Was a p-value < 0.25 again used to declare significance? It helpful to provide the median and IQR for the change parameter you are using.

Results Line 170 - 172 Patients who did not relapse within 24 months after treatment initiation (Responders, R) and patients who had progressed during this period (non-Responders, NR) for immune and non-immune parameters using bulk RNAseq data. Definition of responders and non-responder is problematic given patients (receive pre-operative treatment, surgery, and then post-operative treatment) may have gone off treatment due to toxicity, refusal, or other reasons and then receive alternative therapy. If they are dropped from the analysis, then a bias has been introduced and if not, it is unclear whether the impact of exposure to drugs may have on the endpoint. It is unclear why this dichotomization was done when other aspects of the study looked

a PFS. It is suggested that complete resection rates, the primary endpoint of the clinical trial, be assessed in this manner instead. It is unclear how stable the logistic regression model is given the small number of events per treatment arm.

It is unclear how stable models of PFS with 3 parameters and interaction terms with ~ 20 events (taken from clinical trial manuscript -- unclear how many are in the subset with tumor tissue) in carboplatin + paclitaxel arm and ~32 in the combination + PEM are. It becomes more problematic when examining OS with fewer deaths reported.

Reviewer #4:

Remarks to the Author:

In this manuscript, Olivia Le Saux et al. designed the NeoPembrOv randomized phase II trial and related RNA-seq and multiplexed immunofluorescence staining to decipher the tumor microenvironment between NACT and NACT+P group in high grade serous carcinoma. This idea is interesting and important for clinical treatment. However, this manuscript is preliminary and further clarification and rewriting of the text is needed.

The following are major comments for the authors on the study:

1. As shown in Supplementary Figure 1, there is no difference in infiltrated cells between NACT and NACT+P groups. $P < 0.25$ cannot be considered statistical significance.
2. In line 103, "PTPRC (CD45) and EPCAM expression varied in opposite ways after treatment (Fig. 1A, Supp Fig. 1A)", the author should better use the paired-boxplot and corresponding statistic test to show these findings. Because we cannot find any opposite ways of CD45 after treatment?
3. In Figure 1B, the Log fold change of MCP counter gene expression signature scores between post- vs pre-treatment scores in patients was only single number, how to display it using boxplot with median and quartiles? The confusing problem also occurs in Figures 2D and 2F.
4. The immunosuppressive genes such as TIGIT, LAG3 and PDCD1LG2 are also up-regulated in post-treatment samples, which is inconsistent with the described findings "NACT+P boosted T and B cell mediated immune response" (Supplementary Figure 1B).
5. There is no survival analysis between NACT and NACT+P groups. I want to know whether different treatment options will affect the prognosis of patients?
6. The authors initially compared the difference between NACT and NACT+P, but did not analyze the difference between the different treatment strategies in terms of treatment response. This raises questions about whether there were pre-existing differences between non-responders to NACT and responders to NACT+P prior to treatment? A joint analysis could better identify which patients may benefit more from NACT+P over NACT alone?
7. There are numerous problems with the figures and tables in the manuscript, such as: (1) words of different sizes, colors, and types; (2) some of the border colors of the graph are black and some are gray; (3) the Figure numbers are obscured (Figure 2C, E); (4) the legend contradicts the image presentation, which confuses the reader.

Point-by-point response to the reviewers comments

NCOMMS-23-08444-T

Immunogenomic profiling of the randomized NeoPembrOv trial reveals regulatory T cells and the VEGFR2 angiogenic axis as promising actionable targets to overcome immunoresistance of high-grade ovarian carcinomas

We would like to thank the journal's editors and the reviewers for their useful and constructive comments and suggestions on our manuscript. We provide hereafter a point-by-point response to all comments and submit a revised version of the manuscript, in which all changes appear in the track mode.

Reviewer #1 - Ovarian Cancer, immunotherapy, clinical trials (Reviewer #1 in NCOMMS-23-08443-T) (Remarks to the Author):

Thank you for the opportunity to review this interesting work looking at tumor-based biomarkers of benefit to carboplatin/paclitaxel with or without pembrolizumab in patients with newly diagnosed HGSOC. My comments are as below:

1) *Please discuss the limitations of the study in the discussion section, including the fact that only 64 of 91 patients had sufficient paired tissue for analysis mIF and only 53 for RNAseq. This may have introduced bias into your results. In the supplemental, it would be helpful to have venn diagram of how many samples were obtained from each arm of the study.*

As suggested, in the revised version of the manuscript we discuss the limitations of the present study in a section of the discussion (page 18).

“Some limitations of the present study include the relatively small number of patients included in the phase II trial (n = 91), insufficient quality of certain tumor samples for translational analyses and the absence of samples for some patients, either because they were not eligible for IDA (n = 4) or tumor resection was not possible (CC-3, n = 19).”

As requested by reviewer#3, we also provide a flow diagram depicting the number of samples used for each type of analysis (now **Supp. Fig. 1**) and a supplementary table (**Supp. Table 1**) listing the clinical characteristics of the patients used for mIF analysis vs RNAseq analysis.

2) *For R/NR, how was the 24 month PFS mark selected as clinical endpoint? In the clinical paper, the median duration of follow up was 22 months.*

We recognize that there was a discrepancy between the clinical paper (which was based on the initial clinical data) and the translational paper (which was based on the updated clinical data). We have thus homogenized the two papers.

We used the 24-month PFS mark for the following reasons.

- We did not use the clinical primary endpoint (completeness of cytoreduction score, CC-score) as a landmark to define response as survival endpoints capture responses with immunotherapeutic agents better than the objective response rate or similar parameters such as CC-score (PMID: 19934295, PMID: 26567357). Overall Survival (OS) would have been an alternative, but OS data did not reach maturity due to a significant number of censorings before the median. PFS was therefore considered to be the best endpoint to measure response to Pembrolizumab.

- The median follow-up is 52.4 months with a minimum of 24.6 months and a maximum of 62.7 months. Based on this range, a 24-month cut-off allowed us to have the information (progression/non-progression) for each patient.
- The duration of treatment under pembrolizumab was 24 months. Using this landmark, we could compare progressors vs non-progressors under Pembrolizumab. Besides, patients receiving 2 years of Pembrolizumab were reported to exhibit durable responses in lung cancer justifying this cut-off (PMID: 32078391, PMID: 33872070).
- The 24-month PFS mark compares favorably with median PFS in the ANTHALYA neoadjuvant trial with bevacizumab conducted in French centers (23.5 months, [18.5, 30.6], DOI: 10.1200/JCO.2017.35.15_suppl.5538 Journal of Clinical Oncology 35, no. 15_suppl (May 20, 2017) 5538-5538.)

In the revised version of the manuscript, we have added a justification for the 24-month PFS mark in the results section page 9. *“As patients generally receive Pembrolizumab for a duration of 24 months, we investigated the number of progressors (P) vs non progressors (NP) under Pembrolizumab at 24 months. This landmark is also consistent with the median PFS reported in the French Anthalya trial. This criterion was favored over the completeness of cytoreduction score (CC-score) at interval debulking surgery, as survival endpoints capture responses to immunotherapeutic agents better than the objective response rate or equivalent measures such as the CC-score.”*

We also replaced the terms “response” by “non-progression” under NACT+P throughout the manuscript.

3) *It's also important to note that the use of bevacizumab may also impact your clinical endpoint as bevacizumab has been shown to prolong PFS (although not OS). It would be helpful for the authors to include what percentage of R/NR got bevacizumab, and is it controlled across the two arms? Was bev use incorporated in the statistical models for the predictive biomarkers?*

Bevacizumab was prescribed in the clinical practice after surgery when randomization was already done. Here, we did not include this factor in the statistical model as too few patients did not receive bevacizumab. Yet, we specifically analyzed patients who received bevacizumab to identify if this was a confounding factor. These data are now shown in the new supplementary figure 6B. The number of patients that actually received bevacizumab was not significantly different between arms, and the vast majority of patients received bevacizumab. Nonetheless, we included two sentences in the result section: *“As the fraction of patients that did not receive Bevacizumab was higher in the NACT+P arm compared to the NACT arm, albeit not reaching statistical significance, (14.8% vs 6.67% respectively, $p = 0.328$), we ran the same analysis exclusively on patients who had received the antiangiogenic” and “Moreover, no difference was observed between progressors and non-progressors with regards to the use of Bevacizumab (87.0% vs 91.7% respectively, $p = 0.734$).”*

4) *Why was CK selected as your marker for tumor cells for mIF as opposed to PAX8?*

Pan-CK and Pax8 are both commonly used markers to identify tumor cells in HGSC. Yet, because PAX8 is only expressed in 85 to 95% of HGSC tumor cells, we privileged a pan-CK staining, which should result in fewer false negatives.

5) *In the discussion, the authors should acknowledge that regulatory T cells, as delineated by FOXP3 positivity, have previously been shown to be a negative predictive biomarker for IO*

response and this is not a novel finding. They might also discuss some of the clinical data for t reg depletion strategies, although they have not been successful thus far.

As suggested, we have added a sentence in the discussion section “Even though the presence of regulatory T cells has already been reported to be associated with resistance to ICI, strategies aiming at depleting regulatory T cells using anti-CTLA-4 antibodies have not been successful in HGSC (Zamarin, D. et al., 2020 and Leary, A. et al., 2021).”

6) *It would be helpful to understand the data for the CD8/CD39 predictive ratios a little bit better, as this is somewhat counterintuitive. Prior data have demonstrated that Tex are the cells that are reinvigorated by PD-1/L1 blockade, and prior work suggests that CD39 is a marker of T cell exhaustion (PMID: 29066514). Were other T cell exhaustion markers upregulated in the RNA seq data in the patients who were responders? or regulatory T cell markers? Overall, this is an area that needs some attention and fleshing out to understand potential mechanisms.*

We acknowledge that the interpretation of the CD8/CD39 expression ratio was poorly explained in the submitted manuscript. In the revised version, we provide flow cytometry data obtained with fresh ovarian tumors (n=6, supp Figure 5E) showing that CD39 is expressed at far higher levels on tumor infiltrating Tregs compared to CD4 and CD8 T effector cells. This suggests that the former cells may largely account for CD39 mRNA expression in tumors and that the CD8/CD39 expression ratio may reflect the proportion of CD8 T cells to Tregs. This new piece of evidence was generated by Justine Cinier and Christine Caux, who have therefore been added as co-authors.

7) *The nature of the color coding in Figure 1 panel C is unclear to me - why are some of the bars red, blue and purple?*

*A color code is used to better visualize the pathways common to both arms and those specific of the NACT+P arm. This is now clearly indicated in the legend of **Fig. 1C**.*

8) *Please use asterisks across all the figures to denote statistical significance.*

All figures have been modified in this way.

9) *Figure 2 Panel B and E don't add much to the scientific story and could probably be removed for brevity and clarity.*

***Fig. 2B** was moved to the supplementary data and **Fig 2E** was removed.*

10) *In figure 5, the PFS and OS lines are all overlying because the figure is so small; it would improve legibility if these are made bigger.*

We have modified the graphs of figure 5 and most of the figures to improve their legibility.

Reviewer #2 - Ovarian Cancer, immunotherapy, clinical trials (Reviewer #2 in NCOMMS-23-08443-T) (Remarks to the Author):

The investigation by Le Saux and colleagues used paired samples from the NeoPembrOv randomized phase II trial for RNA-seq and multiplexed immunofluorescence to explore the impact of neoadjuvant chemotherapy (NACT) ± Pembrolizumab (P) on the tumor environment and identify biomarkers for response. The strength of this study is the use of biospecimens from the above noted randomized clinical trial. Unfortunately, some of the methods used make interpretation of the results difficult. Please see below. Critiques

1. It is difficult to evaluate the results in their current form due to the inappropriately low statistical significance cutoff ($p < 0.05$) used for the primary RNAseq analysis causing significant likelihood of false positive results due to multiple comparisons. The pathway enrichment analysis (Fig 1C) does appear to be statistically significant (with adjusted p values), however this analysis is based on the primary DEG from the RNAseq.

To identify differentially expressed genes, we applied a strategy commonly used in several studies. Only genes with a fold-change > 1.5 and a Benjamini and Hochberg adjusted p-value for multiple testing < 0.05 were considered to declare genes to be differentially expressed. The following sentence was modified in the methods section. “As already published, only genes with a fold-change > 1.5 and a Benjamini and Hochberg adjusted p-value for multiple testing < 0.05 were considered.”

Raouf A, et al. Transcriptome analysis of the normal human mammary cell commitment and differentiation process. *Cell Stem Cell*. 2008;3:109–118

Peart MJ, et al. Identification and functional significance of genes regulated by structurally different histone deacetylase inhibitors. *Proc. Natl Acad. Sci. USA*. 2005;102:3697–3702

2. How was tumor percentage controlled for in the RNA seq analyses between pre and post NACT samples? It would be expected that the post samples may have lower tumor percentage which could impact DEG results.

Tumor purity was estimated by the pathologist based on HES-colored slides as well as by using the estimate algorithm based on bulk RNAseq data. Both metrics were correlated ($R=0.71$, $p=3.4 \cdot 10^{-9}$, spearman correlation). As can be seen in Editorial Fig. 1 (below), tumor purity was similar before treatment in both arms:

NACT arm, before treatment median and [1st Qu. - 3rd Qu.] = 0.69 [0.60 - 0.87]

NACT+P arm, before treatment median and [1st Qu. - 3rd Qu.] = 0.77 [0.66 - 0.85]

Wilcoxon test: Tumour purity Before NACT Arm vs. NACT+P Arm = 0.3, 95% CI : [-0.14 - 0.05]

Tumor purity decreased after chemotherapy in both arms, as expected:

NACT arm, after treatment, median and [1st Qu. - 3rd Qu.] = 0.60 [0.48 - 0.73]

NACT+P arm, after treatment, median and [1st Qu. - 3rd Qu.] = 0.66 [0.57 - 0.79]

We acknowledge that this may introduce some bias in the RNAseq analysis. Yet, tumor purity after treatment was similar in both arms (Wilcoxon test: Tumour purity Post Arm A vs. Arm B = 0.18 et 95% CI: [-0.16 - 0.03]), and, to our knowledge, there is so far no approved method to normalize data based on tumor purity. In addition, the fact that a similar increase of T cells is seen both using RNAseq data and multi-IF tissue imaging data argues for the validity of our comparative analysis of RNAseq data. Finally, it has to be noted that our final prediction model

includes only variables measured before treatment, and is thus not influenced by the therapy-induced decrease of tumor cellularity.

Editorial Figure 1. Tumor purity assessed on bulk RNAseq data by using the estimate algorithm in arm A (NACT) vs arm B (NACT+P) before (top) and after treatment (bottom).

As we compare each arm with each other and as we see that tumor purity is similar in both arms at BL and at IDS, we believe that there is no bias in our conclusions.

The former figure could be redesigned and added as a supplementary figure and discussed upon request.

3. The authors should discuss how interaction between their pre-treatment biomarkers (figure 3C). It seems that CD8B/FOXP3 ratio and endothelial cell signature showed a significant interaction with treatment, suggesting that they were predictive biomarkers of response, while the monocyte signature did not. However, the prediction model used all three. It would be helpful to explain the rationale.

Since the experimental arm is not an immune checkpoint inhibitor-only arm, we included parameters that interacted with the experimental arm pembrolizumab+NACT as well as with the NACT only arm. We added the following sentence in the results section “Since the experimental arm is not an ICI-only arm, we included in the model parameters that interact with the treatment and are thus linked to pembrolizumab. We also considered parameters without interaction that are related to NACT.”

4. Manuscript would benefit from scientific editorial review to enhance comprehensibility (e.g. “To precise the predictive impact of the endothelial cell gene signature variable, we categorized patients into “high” and “low” expressors using the optimal threshold, which was close to the median, with 62.3% vs 37.7% of patients having a high or low gene signature expression score, respectively.”)

The manuscript has been reviewed by a scientific writer (B. Manship in the acknowledgment section) and modified in several locations to enhance comprehension as suggested. For example, the sentence cited by the reviewer was modified as follows “*To evaluate the predictive impact of this endothelial cell gene signature variable, we categorized patients into “high” and “low” expressors using the optimal threshold, resulting in 62.3% vs 37.7% of patients having a high or low gene signature expression score, respectively.*”

5. The authors should temper the language in the discussion to reflected the correlative nature of their observations instead of implying causation. For example, in the Discussion they make the statement “Using this multi-omics approach, we found found that Pembrolizumab induced the recruitment of intra-epithelial CD8+PD-1+ T cells.” In fact, their data does not support this assertion. There are multiple other examples in this type of over-stating conclusions in the Discussion section.

In the revised version of the manuscript, we have tempered some of our conclusions, as rightly requested by the reviewer. For example, we modified the sentence cited by the reviewer as follows “*Using this multi-omics approach, we confirmed that chemotherapy remodeled the tumor microenvironment by increasing tumor infiltration by lymphocytes found and showed that the addition of Pembrolizumab amplified these effects with signs of T cells reinvigoration and increased induced the recruitment of intra-epithelial CD8+PD-1+ T cells in the vicinity of tumor cells*”.

Reviewer #3 - Biostatistics, clinical trial design (Reviewer #3 in NCOMMS-23-08443-T) (Remarks to the Author):

A remark diagram should be provided to display how many patients were randomized to each treatment arm and then how many patients had sufficient tumor for determination of immune cell populations in each arm.

As suggested, we have added a flow diagram (**Supp Fig 1**) depicting the number of randomized patients and the number of samples that were used for the different types of analyses. We also now provide a supplementary table (**Supp Table 1**) comparing the clinical characteristics of patients of the overall cohort vs those that contributed material for mIF analysis and/or RNAseq analysis. Emilie Thomas, a biostatistician from from CRCL Gilles Thomas bioinformatic platform was consulted for various biostatistics aspects of the manuscript and was therefore added as a co-author.

Results Line 97 State how many of the 64 patients were assigned to NACT and how many were assigned to NACT + PEM.

As requested, we have added such piece of information in the revised manuscript “*To quantitatively characterize changes in the tumor immune microenvironment induced by neoadjuvant platinum-based chemotherapy ± P, we compared pre- and post-treatment matched tissue samples from 64 patients (see Methods; n = 23 in the NACT arm and n = 41 in the NACT+P arm) with contributive material using bulk RNA-sequencing (RNA-seq) (Supp Table 1 and Supp Fig 1).*”

Results Line 107 and Figure 1 B, C: It is unclear why p-value < 0.25 was chosen as a cutoff point for declaring significance. Sixteen comparisons are being reported in Figure 1B and 12 comparisons are being reported in Supplementary Figure 1C. Instead of loosening the threshold for declaring significance, the p-value should be adjusted to control the false positive rate. Thus, none of the comparisons discussed in lines 94-119 are significant. Any percentages reported such as the percentage of patients with increased expression of the CD8+ T cell gene signature post-treatment should be accompanied by a 95% confidence interval so that the reader can see the variability in that estimate.

Concerning the p-values <0.25, we did not intend to consider them as significant. To avoid this misunderstanding, Figure 1 has been modified and p-values <0.25 are no longer discussed. In the revised version of the manuscript, we still report unadjusted p values (but it is clearly stated in the text) and have added FDR. 95% confidence intervals were also added throughout the results section. The present study, based on a phase 2 trial with a relatively small number of patients, aimed at generating new hypotheses. Due to a small sample size, the study does not have the statistical power to address the multiplication of test issue. Indeed, statistically significant findings are harder to detect with small sample sizes and limit type I errors. We have added in the manuscript that the p values are unadjusted for more clarity (p_{unadj}). Our findings will require validation in other cohorts of ovarian cancer patients treated with immunotherapy with RNAseq data but there is, to our knowledge, no such public dataset yet available. Nevertheless, the impact of the endothelial signature, especially of the expression of VEGFR2/KDR, highlighted in this study, has been validated in another type of cancer using public data from a study evaluating head and neck cancer patients treated with checkpoint inhibitors (PMID: 36038492). This external validation can be added to the manuscript on request (cf editorial figure 2 below).

In the revised version of the manuscript, we now described these limitations as follows (page 18): “Some limitations of the present study include the relatively small number of patients included in the phase II trial ($n = 91$), insufficient quality of certain tumor samples for translational analyses and the absence of samples for some patients, either because they were not eligible for IDS ($n = 4$) or tumor resection was not possible (CC-3, $n = 19$). Nevertheless, this study highlights new potential biomarkers and generates several hypotheses that warrant to be confirmed and further explored in an external cohort of HGSC patients treated with ICI and with public transcriptomic data.”

Editorial Fig 2. OS curves of head and neck cancer patients treated with ICI according to high (red) vs low (orange) expression of *KDR*. *KDR* was categorized using the best cut-off.

Results Line 136-166. It is not clear how changes in density pre and post treatment were analyzed. Was the percent change, fold change or difference examined? What hypothesis test was used to assess changes. Was a p -value < 0.25 again used to declare significance? It helpful to provide the median and IQR for the change parameter you are using.

For changes in cell density between pre and post treatment samples, we compared pre vs post treatment densities using Wilcoxon tests. This is now clearly mentioned in the figure legend of the revised manuscript (Statistical significance was evaluated using Wilcoxon tests). An unadjusted p -value < 0.05 was used to define significance. For $CD8^+PD-1^+$ cells, we also calculated the ratio of cell density in the intra epithelial compartment / stromal compartment before vs after treatment for both treatment arms. The following sentence was modified in the results section “Although the ratio of $CD8^+PD-1^+$ T cell density between tumor and stromal zones was similar between both arms before treatment, it significantly increased in the NACT+P arm after treatment”

Results Line 170 - 172 Patients who did not relapse within 24 months after treatment initiation (Responders, R) and patients who had progressed during this period (non-Responders, NR) for immune and non-immune parameters using bulk RNAseq data. Definition of responders and non-responder is problematic given patients (receive pre-operative treatment, surgery, and then post-operative treatment) may have gone off treatment due to toxicity, refusal, or other reasons and then receive alternative therapy. If they are dropped from the analysis, then a bias has been introduced and if not, it is unclear whether the impact of exposure to drugs may have on the endpoint. It is unclear why this dichotomization was done when other aspects of the study looked a PFS. It is suggested that complete resection rates, the primary endpoint of the clinical trial, be assessed in this manner instead. It is unclear how stable the logistic regression model is given the small number of events per treatment arm.

No patient was removed from the analysis as the minimal follow-up was 24.6 months. No differences in terms of treatment exposure was observed between progressors and non-progressors. This table could be added in the supplementary data upon request (cf **Editorial Table 1** below).

We used the 24-month PFS mark for the following reasons.

- We did not use the clinical primary endpoint (completeness of cytoreduction score, CC-score) as a landmark to define response as survival endpoints capture responses with immunotherapeutic agents better than objective response rate or equivalent such as CC-score (PMID: 19934295, PMID: 26567357). Overall Survival (OS) would have been an alternative, but OS data did not reach maturity due to a significant number of censorings before the median. PFS was therefore considered to be the best endpoint to measure response to Pembrolizumab.
- The median follow-up is 52.4 months with a minimum of 24.6 months and a maximum of 62.7 months. Based on this range, 24 months cut-off allowed us to have the information (progression/non-progression) for each patient.
- The duration of maintenance treatment under pembrolizumab was 24 months. Using this landmark, we can compare progressors vs non-progressors under Pembrolizumab. Besides, patients receiving 2 years of Pembrolizumab were reported to exhibit durable responses in lung cancer justifying this cut-off (PMID: 32078391, PMID: 33872070).
- The 24-month PFS mark compared favorably with median PFS in the ANTHALYA neoadjuvant trial with bevacizumab conducted in French centers (23.5 months, [18.5, 30.6], DOI: 10.1200/JCO.2017.35.15_suppl.5538 Journal of Clinical Oncology 35, no. 15_suppl (May 20, 2017) 5538-5538.)

In the revised version of the manuscript, we have added a justification for the 24-month PFS mark in the results section page 9. *“As patients generally receive Pembrolizumab for a duration of 24 months, we investigated the number of progressors (P) vs non progressors (NP) under Pembrolizumab at 24 months. This landmark is also consistent with the median PFS reported in the French Anthalya trial. This criterion was favored over the completeness of cytoreduction score (CC-score) at interval debulking surgery, as survival endpoints capture responses to immunotherapeutic agents better than the objective response rate or equivalent measures such as the CC-score.”*

We have changed the “responder” and “non-responder” status into “progressive disease” versus “non-progressive disease” under Pembrolizumab to be consistent with the criteria chosen for response evaluation.

Table. Treatment exposure

Exposure	P (n = 54)	NP (n = 35)
Neoadjuvant phase		
Median no. of cycles (range)		
Carboplatin	4 (2–8)	4 (3–6)
Paclitaxel	4 (1–8)	4 (3–6)
Pembrolizumab	4 (1-8)	4 (3–6)
Dose reduction or omission for AE		
Carboplatin	3 (6)	1 (3)
Paclitaxel	5 (10)	4 (11)
Pembrolizumab	0	0
Early discontinuation (<4 cycles) for toxicity, progression, or death		
Carboplatin	1 (2)	0
Paclitaxel	2 (4)	0
Pembrolizumab	1/37 (3)	0
Adjuvant therapy		
Median no. of cycles (range)		
Carboplatin	3 (0–5)	2 (0–6)
Paclitaxel	3 (0–5)	2 (0–6)
Pembrolizumab	14 (0-24)	19 (0–22)
Bevacizumab	14.5 (0–24)	17 (0–22)
Early discontinuation		
Carboplatin	6 (11)	4 (12)
Progression	4 (7)	2 (6)
AE	1 (2)	0
Other	1 (2)	2 (6)
Paclitaxel	11 (20)	5 (14)
Progression	5 (9)	2 (6)
AE	5 (9)	1 (3)
Other	1 (2)	2 (6)
Pembrolizumab	28/37 (76)	11/23 (48)
Progression	19/37 (51)	4/23 (17)
Toxicity	8/37 (22)	5/23 (22)
Other	1/37 (3)	2/23 (9)
Bevacizumab	41 (76)	19 (54)
Progression	26 (48)	11 (31)
Toxicity	11 (20)	3 (9)
Other	4 (7)	5 (14)

Editorial Table 1. Comparison of treatment exposure between progressors and non-progressors.

It is unclear how stable models of PFS with 3 parameters and interaction terms with ~ 20 events (taken from clinical trial manuscript -- unclear how many are in the subset with tumor tissue) in carboplatin + paclitaxel arm and ~32 in the combination + PEM are. It becomes more problematic when examining OS with fewer deaths reported.

Three interaction models have been tested, one for each analyzed parameter. There are 45 events for progression in patients with RNAseq data, so we followed the rule of thumb of one parameter per 10 events.

In order to consider the low number of events and the high number of parameters, we favored the use of a LASSO model over a multiple logistic regression model as the objective was to select useful, and not redundant, features, in a situation where the number of predictor variables was very large with few events and where there was multicollinearity (Tibshirani, R. Regression Shrinkage and Selection via the Lasso. *Journal of the Royal Statistical Society. Series B (Methodological)* 58, 267–288 (1996)).

To limit overinterpretation and overfitting, 100 bootstrap samples were generated and only variables selected through all the 100 corresponding lasso Cox models were retained in the final model. This was added in the results section (p. 10) and methods (p. 24).

Reviewer #4 - Ovarian cancer, immunogenomics (Remarks to the Author):

In this manuscript, Olivia Le Saux et al. designed the NeoPembrOv randomized phase II trial and related RNA-seq and multiplexed immunofluorescence staining to decipher the tumor microenvironment between NACT and NACT+P group in high grade serous carcinoma. This idea is interesting and important for clinical treatment. However, this manuscript is preliminary and further clarification and rewriting of the text is needed. The following are major comments for the authors on the study:

1. *As shown in Supplementary Figure 1, there is no difference in infiltrated cells between NACT and NACT+P groups. $P < 0.25$ cannot be considered statistical significance.*

Concerning the p-values < 0.25 , we did not intend to consider them as significant. To avoid this misunderstanding, Figure 1 has been modified and p-values < 0.25 are no longer discussed. In the revised version of the manuscript, we still report unadjusted p values (but it is clearly stated in the text) and have added FDR.

2. *In line 103, “PTPRC (CD45) and EPCAM expression varied in opposite ways after treatment (Fig. 1A, Supp Fig. 1A)”, the author should better use the paired-boxplot and corresponding statistic test to show these findings. Because we cannot find any opposite ways of CD45 after treatment?*

We modified the sentence as follows to be more precise. “PTPRC (CD45) expression increased significantly in approximately 65%, 95% CI [45-80] of patients in the experimental arm (NACT+P) vs 43%, 95% CI [23-66] (not statistically significant) in the NACT control arm (Fig. 1A-B), indicating that immune cell infiltration preferentially increased when patients received the combination therapy. On the opposite, EPCAM expression decreased significantly after treatment in both arms (Fig. 1A-B).” We also added boxplot to better illustrate those changes (Fig. 1B et Supp Fig. 2A).

3. *In Figure 1B, the Log fold change of MCP counter gene expression signature scores between post- vs pre-treatment scores in patients was only single number, how to display it using boxplot with median and quartiles? The confusing problem also occurs in Figures 2D and 2F.*

To avoid this misunderstanding, we now represent the evolution of signatures pre and post-treatment. Fig 1B was therefore modified. We did the same for density (figure 2). For current supp Fig. 3C, we clarified it as the ratio between the post-treatment and the pre-treatment nearest neighbor distance for each patient.

4. *The immunosuppressive genes such as TIGIT, LAG3 and PDCD1LG2 are also up-regulated in post-treatment samples, which is inconsistent with the described findings “NACT+P boosted T and B cell mediated immune response” (Supplementary Figure 1B).*

We do not think that this is inconsistent as immunosuppressive molecules such as TIGIT, LAG3 and PDCD1LG2 can also be induced/upregulated on T cells or B cells. For example, LAG3 transcription is upregulated following activation through the T cell Receptor (TCR) (PMID: 33488626) and similar results were reported for TIGIT (PMID: 37756440) and PDCD1LG2 (PMID: 35276636). Besides, an increase in suppressive cells can also accompany induction of effector immune responses.

5. There is no survival analysis between NACT and NACT+P groups. I want to know whether different treatment options will affect the prognosis of patients?

These data are provided and discussed in the clinical paper. We have added a sentence referring to these data in the revised version of the manuscript, showing no statistically significant difference of survival between the 2 arms (methods section lines 493-497). In spite of a study that failed to show the benefit of adding Pembrolizumab to standard NACT±bevacizumab, we find predictive biomarkers of progression specifically in the experimental arm.

6. The authors initially compared the difference between NACT and NACT+P, but did not analyze the difference between the different treatment strategies in terms of treatment response. This raises questions about whether there were pre-existing differences between non-responders to NACT and responders to NACT+P prior to treatment ? A joint analysis could better identify which patients may benefit more from NACT+P over NACT alone?

We refrained from conducting a joint analysis as we solely considered pre-existing parameters, excluding those evaluated during interval debulking surgery (IDS). Figures linked to parameters evaluated at the IDS were eliminated from the corresponding **Supplementary Figure**. Additionally, we chose to classify responses into progression and non-progression at the 24-month timepoint, following the rationale outlined for R1 and R3, thus not considering PFS as a continuous outcome.

To determine if parameters were associated with response to NACT+P and NACT, we used interaction tests as it is the gold standard to identify predictive biomarkers of response to treatment (DOI: 10.1200/JCO.2015.63.3651 Journal of Clinical Oncology 33, no. 33 (November 20, 2015) 3968-3971.).

7. There are numerous problems with the figures and tables in the manuscript, such as: (1) words of different sizes, colors, and types; (2) some of the border colors of the graph are black and some are gray; (3) the Figure numbers are obscured (Figure 2C, E); (4) the legend contradicts the image presentation, which confuses the reader.

We have modified all figures and legends for them to be more homogeneous and legible.

Reviewers' Comments:

Reviewer #1:

Remarks to the Author:

My concerns have been addressed.

Reviewer #2:

Remarks to the Author:

I appreciate the authors addressing reviewers' comments in good faith

Reviewer #4:

Remarks to the Author:

Although the authors have done some analysis to address the comments raised by the reviewers, there are still serious concerns as follows:

1. The lack of immunogenomic data is notable. The study relies exclusively on transcriptomic data obtained from RNA sequencing of tumor samples from patients with high-grade ovarian cancer. This raises questions about the potential overestimation of the study results.
2. An anomaly is observed for CD8B/FOXP3 in NACT + P, which is evident in Figure 1A but conspicuously absent in the corresponding box plot in Figure 1B. This discrepancy is cause for concern. The rationale for the disproportionate focus on CD8B/FOXP3 and the omission of CD8A/FOXP3 should be clarified.
3. In line 291, the statement "Importantly, high expression of VEGFR2 on CD31+ endothelial cells was associated with increased survival of patients after treatment with NACT+P (Fig. 6D)" is not supported by a review of Figure 6D. The presentation of Figure 6D lacks clarity and is confusing to the reader.
4. In the rebuttal text Editorial Fig. 2, careful review is required to confirm the accuracy of the "time in days" description. The original manuscript uses the description "time in months".
5. Address irregularities in statistical testing:
 - a. For multiple repeatability tests, correction of the P value is mandatory to reduce Type I errors. If no correction has been applied, a detailed explanation, supported by evidence, must be provided.
 - b. It is surprising that $-\log_{10}(p.\text{adj})$ is less than 0. However, $-\log_{10}(p.\text{adj}) = -50$, which means that $p.\text{adj}$ is greater than 10 billion. This is impossible!

Reviewer #5:

Remarks to the Author:

Thanks to the authors for providing detailed response. Further clarifications are needed for a few analysis.

1. Please clarify and be precise on the methods of Wilcoxon test that were used to compare biomarker levels. Two different types of Wilcoxon tests were required depending on whether it was the comparison of pre- and post-treatment difference with paired samples, or between independent group comparisons. Wilcoxon signed-rank test should be used for paired-sample comparisons to take into account the correlations of the paired samples, while Wilcoxon rank sum test or Mann-Whitney test should be used to compare independent groups.
2. The development of multivariable model to predict response needs further clarification (Line 196-214). Were the data of both arms used for model selection using LASSO method, or it only used the data of NACT+P arm? If it was former, was the interaction term of the level of each marker by treatment arm considered in model selection? If yes, and the final model included the interactions of marker levels by treatment arm, the parameter beta for the main effect of the markers can no longer be interpreted as odds ratios. If the data from both arms were used, the interaction terms should be considered for variable selection, as some markers of interest may be

predictive markers .

3. It was unclear what comparison the p value represented on some Kaplan-Meier curves, for instance, Figure 5A and 5C, Figure 6A and 6D, and a few more supplementary figures. Were they log-rank test to compare the four groups, or were they p values for the interactions of marker and treatment arm? It needs to be clarified. It was unclear how meaningful it was to compare the 4 groups with log-rank test if that was the case.

4. In the analysis of Cox model and LASSO selection, the biomarkers were categorized with the optimal cut-off. What was the method used to determine the optimal cut-off thresholds. Please clarify.

Point-by-point response to the reviewers comments

NCOMMS-23-08444-T

Immunogenomic profiling of the randomized NeoPembrOv clinical trial reveals regulatory T cells and the VEGFR2 angiogenic axis as promising actionable targets to overcome immunoresistance of high-grade ovarian carcinomas

We would like to thank the journal's editors and the reviewers for their additional and constructive comments on our manuscript. We provide hereafter a point-by-point response to all comments and submit a revised version of the manuscript, incorporating changes in response to all the comments (all changes appear in the track mode).

REVIEWER COMMENTS

Reviewer #1 (Remarks to the Author):

My concerns have been addressed.

Reviewer #2 (Remarks to the Author):

I appreciate the authors addressing reviewers' comments in good faith

We would like to thank both reviewers for their positive feedback.

Reviewer #4 (Remarks to the Author):

Although the authors have done some analysis to address the comments raised by the reviewers, there are still serious concerns as follows:

1. The lack of immunogenomic data is notable. The study relies exclusively on transcriptomic data obtained from RNA sequencing of tumor samples from patients with high-grade ovarian cancer. This raises questions about the potential overestimation of the study results.

The study, which aimed at exploring the impact of neoadjuvant chemotherapy ± pembrolizumab (P) on the tumor immune environment and at identifying parameters associated with response to immunotherapy, combined both transcriptomic and multi-immunofluorescence data. To address the potential overestimation of the study results, multi-IF data confirmed at the protein level several salient results obtained by RNAseq, including the impact of Pembrolizumab on CD8⁺ T cells and the association between a high VEGFR2 expression and resistance to P (figure 6). We agree that immunogenomic implies DNA sequencing which would provide a genetic characterization of both cancer cells and immune cells. Nonetheless, some also report a broader sense including the description of molecular features and changes in immune cells associated with disease using functional genomics technologies¹ including transcriptomics data. As our study did not include a DNA dimension,

we modified the title to be less misleading. Our new title is “Immunomic profiling of the randomized NeoPembrOv clinical trial reveals regulatory T cells and the VEGFR2 angiogenic axis as promising actionable targets to overcome immunoresistance of high-grade ovarian carcinomas”.

2. An anomaly is observed for CD8B/FOXP3 in NACT + P, which is evident in Figure 1A but conspicuously absent in the corresponding box plot in Figure 1B. This discrepancy is cause for concern. The rationale for the disproportionate focus on CD8B/FOXP3 and the omission of CD8A/FOXP3 should be clarified.

Thank you for pointing this anomaly which was due to an incorrect row scaling in fig. 1A. This has now been fixed and we provide a corrected heatmap new figure 1.

We have decided to focus mainly on *CD8B*, as this gene, and not *CD8A*, is part of the CD8⁺ T cell signature of MCP counter², which consider only specific genes. Indeed, CD8A can be expressed on other immune cells (<https://www.uniprot.org/uniprotkb/P01732/entry#expression>) such as plasmacytoid dendritic cells³, NK cells⁴ and monocytes⁵. In contrast, CD8 beta is more specific of CD8⁺ T cells⁶ (<https://www.uniprot.org/uniprotkb/P10966/entry#expression>). To be transparent, we justify this choice in the methods section “To infer the ratio of CD8⁺ T cells to Tregs, we calculated the *CD8B/FOXP3* expression ratio and privileged *CD8B* over *CD8A* as this gene is more specific of CD8⁺ T cells and is included in the CD8⁺ T lymphocytes signature of MCP counter”.

3. In line 291, the statement "Importantly, high expression of VEGFR2 on CD31+ endothelial cells was associated with increased survival of patients after treatment with NACT+P (Fig. 6D)" is not supported by a review of Figure 6D. The presentation of Figure 6D lacks clarity and is confusing to the reader.

Thank for detecting this error, which has been corrected in the revised version of the manuscript. It is indeed a low expression of VEGFR2 that is associated with increased patient overall survival. To be less confusing to the reader (as also pointed by reviewer#5), we also decided to indicate on the survival curves, the pairwise comparison of survival curves and to remove the global p-value, which can be misleading and is not relevant here.

4. In the rebuttal text Editorial Fig. 2, careful review is required to confirm the accuracy of the "time in days" description. The original manuscript uses the description "time in months".

This is indeed a mistake, the time should have appeared “in months” (see supp. figure 6C).

5. Address irregularities in statistical testing:

a. For multiple repeatability tests, correction of the P value is mandatory to reduce Type I errors. If no correction has been applied, a detailed explanation, supported by evidence, must be provided.

In the revised version of the manuscript, we report unadjusted p values (this is clearly stated in the text by the abbreviation “_{unadj}P”) and we have added FDR in order to present the results and the associated confidence level as honestly as possible. With the same objective of transparency with regard to the results, 95% confidence intervals were also added throughout the results section. The present study, based on a phase 2 trial, aimed at generating new hypotheses. Due to its relatively small sample size, this study does not have sufficient statistical power to address the issue of multiple tests. The analysis reported here is an exploratory endpoint and as such the use of unadjusted p values is acceptable. Other studies, even at a larger scale, reported such unadjusted p values⁷. Indeed, in the PRIMA trial, analyses of secondary endpoints were not

adjusted for multiple comparisons⁷. Similar approaches were used in different papers including two recent papers published in *Nature Communications*^{8,9}. This approach is clearly stated in the methods section “Due to the exploratory nature of the study, unadjP-values < 0.05 were considered significant”.

Nevertheless, the impact of the endothelial signature, especially of the expression of VEGFR2/KDR, highlighted in this study, has been validated in another type of cancer using public data from a study evaluating checkpoint inhibitors in head and neck cancer patients¹⁰. This external validation was added to the manuscript in supp figure 6C.

We are aware that, ideally, our findings will need to be validated in other cohorts of patients with ovarian cancer who have been treated with immunotherapy. However, to our knowledge, no such public RNAseq dataset is yet available.

b. It is surprising that $-\log_{10}(p.\text{adj})$ is less than 0. However, $-\log_{10}(p.\text{adj}) = -50$, which means that $p.\text{adj}$ is greater than 10 billion. This is impossible!

Thank you for pointing this mistake. It is indeed not $-\log_{10}(p.\text{adj})$ values, but scaled values. We have now addressed this issue and corrected the figure 1C. Figure 1C now represents gene ratio and GO-BP pathways as it is frequently depicted.

References

1. Hossain, F., Majumder, S. & Miele, L. Chapter 7 - Immunogenomics: steps toward personalized medicines. in *Clinical Precision Medicine* (ed. Crabtree, J. S.) 73–90 (Academic Press, 2020). doi:10.1016/B978-0-12-819834-6.00007-0.
2. Becht, E. *et al.* Estimating the population abundance of tissue-infiltrating immune and stromal cell populations using gene expression. *Genome Biol.* **17**, 218 (2016).
3. Schuster, P., Thomann, S., Werner, M., Vollmer, J. & Schmidt, B. A subset of human plasmacytoid dendritic cells expresses CD8 α upon exposure to herpes simplex virus type 1. *Front. Microbiol.* **6**, 557 (2015).
4. Addison, E. G. *et al.* Ligation of CD8 α on human natural killer cells prevents activation-induced apoptosis and enhances cytolytic activity. *Immunology* **116**, 354–361 (2005).
5. Gibbings, D. J., Marcet-Palacios, M., Sekar, Y., Ng, M. C. & Befus, A. D. CD8 α is expressed by human monocytes and enhances Fc γ R-dependent responses. *BMC Immunol.* **8**, 12 (2007).
6. DiSanto, J. P., Smith, D., de Bruin, D., Lacy, E. & Flomenberg, N. Transcriptional diversity at the duplicated human CD8 beta loci. *Eur. J. Immunol.* **23**, 320–326 (1993).
7. González-Martín, A. *et al.* Niraparib in Patients with Newly Diagnosed Advanced Ovarian Cancer. *N. Engl. J. Med.* **381**, 2391–2402 (2019).
8. Hong, L. *et al.* Efficacy and clinicogenomic correlates of response to immune checkpoint inhibitors alone or with chemotherapy in non-small cell lung cancer. *Nat. Commun.* **14**, 695 (2023).
9. Taraborrelli, L. *et al.* Tumor-intrinsic expression of the autophagy gene Atg16l1 suppresses anti-tumor immunity in colorectal cancer. *Nat. Commun.* **14**, 5945 (2023).
10. Foy, J.-P. *et al.* Immunologically active phenotype by gene expression profiling is associated with clinical benefit from PD-1/PD-L1 inhibitors in real-world head and neck and lung cancer patients. *Eur. J. Cancer Oxf. Engl. 1990* **174**, 287–298 (2022).

Reviewer #5 (Remarks to the Author):

Thanks to the authors for providing detailed response. Further clarifications are needed for a few analysis.

1. Please clarify and be precise on the methods of Wilcoxon test that were used to compare biomarker levels. Two different types of Wilcoxon tests were required depending on whether it was the comparison of pre- and post-treatment difference with paired samples, or between independent group comparisons. Wilcoxon signed-rank test should be used for paired-sample comparisons to take into account the correlations of the paired samples, while Wilcoxon rank sum test or Mann-Whitney test should be used to compare independent groups.

Thank you for this important comment. Indeed, we used Wilcoxon signed-rank test for paired-sample comparisons and Wilcoxon rank sum test to compare independent groups. The following sentence has been added in the methods section “Wilcoxon signed-rank test was used for paired-sample comparisons, while Wilcoxon rank sum test was used to compare independent groups”. The type of test that was performed is also precised in the figure legends.

2. The development of multivariable model to predict response needs further clarification (Line 196-214). Were the data of both arms used for model selection using LASSO method, or it only used the data of NACT+P arm? If it was former, was the interaction term of the level of each marker by treatment arm considered in model selection? If yes, and the final model included the interactions of marker levels by treatment arm, the parameter beta for the main effect of the markers can no longer be interpreted as odds ratios. If the data from both arms were used, the interaction terms should be considered for variable selection, as some markers of interest may be predictive markers.

Thank you for this interesting comment. We indeed used the data of only the NACT+P arm. This was precised in the text “To select significant and independent biomarkers associated with response to NACT+P, we used a multivariate logistic regression model with least absolute shrinkage and selection operator (LASSO) on patients treated with NACT+P. We included in the model all pre-treatment biomarkers significantly associated with response to NACT+P in univariate analyses...”.

3. It was unclear what comparison the p value represented on some Kaplan-Meier curves, for instance, Figure 5A and 5C, Figure 6A and 6D, and a few more supplementary figures. Were they log-rank test to compare the four groups, or were they p values for the interactions of marker and treatment arm? It needs to be clarified. It was unclear how meaningful it was to compare the 4 groups with log-rank test if that was the case.

Indeed, we had indicated the global p-value to compare the 4 groups using the likelihood ratio (LR) test, as it has a better behavior for small sample sizes and is generally preferred in this setting. As comparing four groups together was not related to the conclusions we raised, we replaced it by pairwise comparisons between survival curves. Such hazard ratios and corresponding p-values are now indicated in Figures 5A, 5C, 6A, 6D and in Supp Figures 5B, 5F, 5G, 6A, 6D. Regarding the use of the LR test, we added the following sentence in the methods section “*Comparison of survival curves was performed using the likelihood ratio test due to sample size*”.

4. In the analysis of Cox model and LASSO selection, the biomarkers were categorized with the optimal cut-off. What was the method used to determine the optimal cut-off thresholds. Please clarify.

We used the `surv_cutpoint` function to categorize variables with the optimal threshold (survminer package). The survminer package determines the optimal cutpoint for continuous variables using the maximally selected rank statistics. We added the following sentence in the methods section “*Optimal cutoffs for continuous exploratory variables scores were chosen based on a maximally selected rank statistics using survminer (v0.4.9).*”

Reviewers' Comments:

Reviewer #4:

None

Reviewer #5:

Remarks to the Author:

The authors have addressed my concerns. With my comment #2 regarding model selection and predictive biomarkers, it helps to clarify that after the variable selection based on NACT+P arm, interactions of the selected markers with treatment arms were tested to assess whether they were predictive biomarkers.

Point-by-point response to the reviewers comments

REVIEWERS' COMMENTS

Reviewer #5 (Remarks to the Author):

The authors have addressed my concerns. With my comment #2 regarding model selection and predictive biomarkers, it helps to clarify that after the variable selection based on NACT+P arm, interactions of the selected markers with treatment arms were tested to assess whether they were predictive biomarkers.

We would like to thank reviewer#5 for his/her positive feedback.